# Geometrical Representation of a Polarisation Management Component on a SOI Platform

**DOI:** 10.3390/mi10060364

**Published:** 2019-05-30

**Authors:** Massimo Valerio Preite, Vito Sorianello, Gabriele De Angelis, Marco Romagnoli, Philippe Velha

**Affiliations:** 1Scuola Superiore Sant’Anna—TeCIP Institute, Via Moruzzi 1, 56124 Pisa, Italy; valeriopreite@gmail.com; 2CNIT—Laboratory of Photonic Networks, Via Moruzzi 1, 56124 Pisa, Italy; vito.sorianello@cnit.it (V.S.); gabriele.deangelis@cnit.it (G.D.A.); marco.romagnoli@cnit.it (M.R.)

**Keywords:** Silicon Photonics, off-chip coupling, polarisation controller, integrated polarimeter, polarisation multiplexing, polarisation shift keying

## Abstract

Grating couplers, widely used in Silicon Photonics (SiPho) for fibre-chip coupling are polarisation sensitive components, consequently any polarisation fluctuation from the fibre optical link results in spurious intensity swings. A polarisation management componentis analytically considered, coupled with a geometrical representation based on phasors and Poincaré sphere, generalising and simplifying the treatment and understanding of its functionalities. A specific implementation in SOI is shown both as polarisation compensator and polarisation controller, focusing on the operative principle. Finally, it is demonstrated experimentally that this component can be used as an integrated polarimeter.

## 1. Introduction

Silicon photonics, thanks to its compatibility with CMOS technology, is imposing itself for large scale fabrication of low cost and small footprint photonic integrated circuits (PICs). One of Silicon Photonics main challenges is coupling light in and out from the chip in an efficient and practical way. The high index contrast between silicon and silica enables the use of grating couplers (GC), that, to our knowledge, are the most widespread off chip coupling solution, mainly thanks to the design flexibility deriving from the fact that they can be placed nearly everywhere on the chip and are not constrained to the chip edge. An important limitation of GCs is their polarisation sensitivity; consequently, the input polarisation fluctuations that regularly occur in optical fibres as a result of deformations or temperature changes translate into random spurious amplitude modulations. Other coupling schemes, such as butt or end fire coupling, are possible, but they are not of interest for this paper.

The problem of assuring polarisation tracking was broadly addressed decades ago, as it is critical for the working of coherent optical systems, given the need to match the time varying State of Polarisation (SOP) of the input signal to the local oscillator’s one. The proposed solutions were based on fibre squeezers [1], lithium-niobate integrated devices [2,3] and Planar Lightwave Circuit (PLC) technology [4].

In [5], the proof of concept of Caspers et al. [6] was further developed into a fully functional building block with two independent phase shifters and was integrated at the two ends of the bus waveguide in a subsystem presented in [7], thus making it transparent to polarisation fluctuations. Until now, the strategy to cope with polarisation fluctuations has been to separate the signal in two orthogonal polarisations and duplicate the circuits, in what is called polarisation diversity scheme. In this example, the compensator is able to convert any SOP into the standard TE mode of Silicon Photonics waveguide eluding a duplicated circuit.

The current article extends the results in [5,6], and aims at developing a simple but precise and exhaustive geometrical picture based on both phasor and Bloch sphere representation. The use of this pictorial representation is illustrated by solving the problem of the frequency response together with its effect on the point representing the SOP on the Poincaré sphere.

In polarisation tracking, sometimes it is not enough to have a device that can compensate all possible SOPs, but it may be desirable to have an endless system, i.e., one where resets—when the physical quantity producing the phase shift reaches any end of its range and a phase jump of an integer number of 2π must be applied—are collision-free and do not provoke transmission disruptions nor information losses. Even better is a reset-free system, that is, one in which potentially unlimited phase shifts can be achieved with the control quantity limited in a finite interval. Usual wave plate transformers possess this last property [2] and have been implemented in lithium-niobate [8,9]. Quarter- and half-wave plates can be implemented in PLC [10], thus even in SiPho. Hence, the same functionality of the device in [2] can be achieved in SiPho, at the price of a greater complexity.

This paper is organised in the following way: after a short description of the device structure in Section 1.1, Section 2 analyses single wavelength operation, while Section 3 examines the frequency response. In both cases, both operating modes are analysed: here “compensator” refers to the case in which the light enters from the 2DGC and the circuit acts so that all the power is routed toward one of its two ends, while “controller” means the reciprocal case where the input is one of the two waveguides and the device settings determine the polarisation exiting from the 2DGC.

In Section 2.1, a phasor representation illustrates the particular case of the polarisation compensation. Two classes of periodical solutions are found, for any input SOP. As a corollary, the knowledge of the phase shifts needed to compensate the input of an unknown SOP can be used to measure without ambiguity this SOP. For the first time, it is demonstrated that a polarimeter can be made in a silicon photonics platform.

Next, several properties are derived thanks to the graphical representation as a function of two physical quantities: the applied phase shifts.

In Section 2.2, the operation as polarisation controller is examined, proving that all the SOPs can be generated and the effect of the two phase shifters on the corresponding point on Poincaré sphere is described.

Section 3 examines the frequency response for both the compensator and the generator operation. Using the expression provided in [11] for the frequency and temperature dependence of SOI effective index, an expression for the phase shift frequency dependence is derived. A transcendental implicit equation for the −3 dB bandwidth is found and it is shown that the frequency response is non trivial depending in both the considered input SOP and the chosen phase shift pair. Then, those results are transposed to Poincaré sphere.

Finally, Section 4 presents an experimental test of the derived results.

### 1.1. Device Schematic

The device exploits a 2D grating coupler (2DGC) to split the incoming field into two orthogonal components. These two components are then fed into two distinct integrated waveguides.

The first stage introduces a first phase shift, then the fields in the two branches are combined and again split by means of an MMI, and undergo another phase shift.

Eventually, a second MMI recombines the fields and sends its output to the two exits.

In [7], the top output is terminated on a monitoring photodiode, while the bottom one is connected to the rest of the PIC. The photodiode current is minimised during the module tuning to make sure that all the optical power goes to the bottom output and therefrom to the downhill PIC. For consistency with that article and the work in [5], in the rest of this paper, we consider that only port “B” in Figure 1 is used.

However, that is not the only possible arrangement. On the contrary, each port can be connected to a distinct photonic module, which processes the information encoded on the SOP orthogonal to that of the other output, for instance in a POLarisation Shift Keying (POLSK) scheme.

## 2. Single Wavelength Operation

### 2.1. Compensator

The overall device, excluding the grating couplers, can be viewed as the cascade of four blocks; correspondingly, its total transfer matrix is the product of the four individual blocks matrices, which are phase shifters (PS) and couplers (C), respectively, and read:(1)$$M_{PS}=\left[\begin{array}{cc} e^{i\;\Delta \varphi } & 0\\ 0 & 1 \end{array}\right]\;M_{C}=\left[\begin{array}{cc} \cos(\kappa ) & -i\sin(\kappa )\\ -i\sin(\kappa ) & \cos(\kappa ) \end{array}\right]=\frac{1}{\sqrt{2}}\left[\begin{array}{cc} 1 & -i\\ -i & 1 \end{array}\right]$$
where $\Delta \varphi =\Delta_{tb}\left(\beta L\right)=\Delta_{tb}\left(\beta \right)L$ and the function $\Delta_{tb}$ denotes the difference of the quantity between brackets for the top and bottom arm. It is applied just to the propagation constant β as the two arms are ideally of equal length. The coupling coefficient is assumed to be κ=π/2 (3 dB coupler).

Thus, the overall transfer matrix reads:(2)$$T=M_{C_{2}}M_{PS_{2}}M_{C_{1}}M_{PS_{1}}=\frac{e^{-i\;\psi }}{2}\;\left[\begin{array}{cc} \;e^{i\;\phi_{1}}\left(1-e^{i\;\phi_{2}}\right) & -i\left(1+e^{i\;\phi_{2}}\right)\\ -i\;e^{i\;\phi_{1}}\left(1+e^{i\;\phi_{2}}\right) & -\;\left(1-e^{i\;\phi_{2}}\right) \end{array}\right]$$
where ψ is an arbitrary absolute phase shift, and $\phi_{1}$ and $\phi_{2}$ are the applied phase shifts, as shown in Figure 1. As input, we consider a generic polarisation, described by the Jones vector in three equivalent forms:(3)$$A_{in}=e^{i\;\delta_{1}}\left(\begin{array}{c} a_{1}\;\\ a_{2}\;e^{i\;\delta } \end{array}\right)=e^{i\;\delta_{1}}\left(\begin{array}{c} \cos\alpha \\ \sin\alpha \;e^{i\;\delta } \end{array}\right)=\left(\begin{array}{c} q_{1}\\ q_{2} \end{array}\right)$$
where q_1_ and q_2_ are defined as two complex values representing the input vector polarisation. The main hypothesis of this works is that the 2DGC is supposed to split the TE and TM components without affecting their relative amplitude and phases (attenuating or phase shifting them in the same way, i.e., without crosstalk), so that the input vector to the rest of the circuit can be assumed to coincide with the above Jones vector.

Thus, the output complex amplitudes vector from the circuit is:(4)$$A_{out}=\left(\begin{array}{c} a_{up}\\ a_{bottom} \end{array}\right)=T\;A_{in}=\frac{e^{-i\;\vartheta }}{2}\left(\begin{array}{c} \;a_{1}\left(1-e^{i\;\phi_{2}}\right)-i\;a_{2}\;e^{i\;\left(\delta -\phi_{1}\right)}\left(1+e^{i\;\phi_{2}}\right)\\ -i\;a_{1}\left(1+e^{i\;\phi_{2}}\right)-\;a_{2}\;e^{i\;\left(\delta -\phi_{1}\right)}\left(1-e^{i\;\phi_{2}}\right) \end{array}\right)$$

To send all the power in the bottom waveguide, the condition to be fulfilled is:(5)$$A_{out}=\left(\begin{array}{c} a_{up}\\ a_{bottom} \end{array}\right)=e^{i\theta }\left(\begin{array}{c} 0\\ 1 \end{array}\right)$$

Again, the absolute phase shift term in front of the matrix, $\vartheta =\psi +\delta_{1}+\phi_{1}$, can be ignored, whereas the one in front of the desired output, θ, is kept for the sake of completeness.

#### 2.1.1. Graphical Solutions in the Phasor Space

We introduce a graphical representation based on phasors, commonly used in quantum mechanics, to offer an intuitive view of the components’ behaviour. The two terms of the vector of Equation (4) are represented using the convention that a phase shift $e^{i\varphi }$ corresponds to a counter clockwise rotation of angle φ. The first term of the top part $a_{1}\left(1-e^{i\phi_{2}}\right)$ brings us to point A and the second term $i\;a_{2}\;e^{i\;\left(\delta -\phi_{1}\right)}\left(1+e^{i\;\phi_{2}}\right)$ brings us to point B.

The fruitfulness of this pictorial approach lies in the immediate and simple way to find the solutions to Equation (4), that is, of solving the problem.

In order for the difference appearing in Equation (4) first row to be zero, its two terms must be equal, that is to say, the corresponding phasors tips A and B must coincide in Figure 2a (small red circle).

This can happen only if both tips lie on the intersection between the dashed circumferences (the other intersection, where the phasors “nocks” are, is not a solution, as $\phi_{2}$ should be simultaneously 0 for $a_{1}$ and π for $a_{2}$); this means that the four represented arrows form a closed quadrilateral (Figure 3a).

In such a circumstance, the quadrilateral opposite angles on the tips of $a_{1}$ and $a_{2}\;e^{i(\gamma +\pi /2)}$ are supplementary by construction, and, therefore, such must be the other two angles as well.

Furthermore, as shown in Figure 3a, those angles are congruent, because they are the sum of angles on the base of two isosceles triangles (because they are inscribed in the two circumferences). Thus, they must equal a right angle, in order to be supplementary.

In turn, that imposes the following relationships:(6)$$\delta -\phi_{1}=n\pi \;\mathrm{or}\;\phi_{1}=\delta +n\pi$$

Thus, the phase shift introduced by the first stage must match that between the input Jones vector components, δ modulo π.

The requirement of cancelling the first component of Equation (4) translates into
(7)$$\begin{array}{cc} e^{i\;\phi_{2}}=\frac{a_{1}-i\;a_{2}\;e^{i\delta -\phi_{1}}}{a_{1}+i\;a_{2}\;e^{i\delta -\phi_{1}}} & =\frac{a_{1}\mp i\;a_{2}}{a_{1}\pm i\;a_{2}}=e^{\mp i\;2\arctan(a_{2}/a_{1})}\\ i.e.,\;\phi_{2} & =\mp \;2\arctan(a_{2}/a_{1})=\mp 2\alpha \end{array}$$

This condition can be deduced more easily from Figure 3a, considering the rectangular triangles with catheti of length $a_{1}$ and $a_{2}$. Note that the parity of *n* in Equation (6) determines the sign of the solution for $\phi_{2}$, as well as which pair of diagrams is to be considered in Figure 3.

Now, considering the bottom component, it can be shown to be
(8)$$-i\;e^{-i\;\left(\psi +\delta_{1}+\phi_{1}\right)}\;\left(a_{1}\mp i\;a_{2}\right)$$

Even though the results can be obtained through straightforward calculations, a geometrical resolution based on the phasor representation offers a quicker and more insightful method.

Starting from Figure 2b, the term can be set to zero rotating the phasors such that their tips meet at the intersection of the circles, as in Figure 3a(a’).

When looking at the phasor of the bottom term of Equation (4) when the top term is equal to zero, we obtain the representation of Figure 3b(b’). Dropping the heights (points B and C), it is clear that the length of the bases’ sum is twice the hypotenuse of the rectangle triangle with $a_{1}$ and $a_{2}$ as catheti, as shown in Figure 3b(b’).

This shows that it is possible, at least at one single wavelength, to compensate the polarisation in such a way that the amplitude at one exit port is zero while the other is maximum.

Summarising, the two solutions for the phase shifts to be applied are:
(9a)$$\begin{array}{cccc} \;\phi_{1}^{\prime } & =\delta +2n\;\pi & \;\phi_{2}^{\prime } & =-2\arctan(a_{2}/a_{1})+2m\;\pi \; \end{array}$$
(9b)ϕ1″=δ+(2n+1)πϕ2″=−2arctan(a2/a1)+2mπ

Until now, the Jones representation (**J**) has been used but a more complete approach can be derived using Stokes parameters. The Stokes parameters are defined in terms of *Pauli matrices*
$\sigma_{i}$ as (eq. 2.5.24 [12])
(10)$$s_{i}≐J^{†}\sigma_{i}J$$
with the notation of Equation (3) they become
(11)$$\begin{array}{cccc} s_{0} & =|q_{1}{|}^{2}+{\left|q_{2}\right|}^{2} & =a_{1}^{2}+a_{2}^{2} & =1\\ s_{1} & =|q_{1}{|}^{2}-{\left|q_{2}\right|}^{2} & =a_{1}^{2}-a_{2}^{2} & =\cos(2\alpha )\\ s_{2} & =q_{1}q_{2}^{*}+q_{2}q_{1}^{*} & =2\;a_{1}a_{2}\cos\delta & =\sin(2\alpha )\;\cos\delta \\ s_{3} & =i(q_{1}q_{2}^{*}-q_{2}q_{1}^{*}) & =2\;a_{1}a_{2}\sin\delta & =\sin(2\alpha )\;\sin\delta \end{array}$$
so the quantities appearing in Equation (9) can be expressed as:(12)$$2\alpha =2\arctan\left(a_{2}/a_{1}\right)=\arccos\left(s_{1}\right)\;\delta =\arctan\left(\frac{s_{3}}{s_{2}}\right)$$

One must note that the signs depend on the particular placement of the phase shifters that has been considered. If one shifter is moved to the other arm, this will result in a sign change in the formula above. In addition, using a push–pull configuration would halve the phase shift to be applied on each individual heater. The practical advantage is that, if a negative shift in the (−π,0) interval is to be applied, actually a shift between π and 2π would have to be used in a single heater configuration, whereas in the symmetric case a positive shift between 0 and π would suffice.

#### 2.1.2. Intensity Surface

It is interesting to consider, for a given input polarisation state, how the output intensity depends on the two phase shifts (that are not necessarily set to the values which yield perfect compensation). To do so, we expand the squared modulus of Equation (4) bottom component:(13)$$P_{L}={\left|\frac{1}{2}\left[-i\;a_{1}\left(1+e^{i\;\phi_{2}}\right)-\;a_{2}\;e^{i\;\left(\delta -\phi_{1}\right)}\left(1-e^{i\;\phi_{2}}\right)\right]\right|}^{2}$$

It is convenient to use the form of Equation (3) in terms of the angles α and δ.

After several passages and trigonometric identities, the above formula can be shown to become:(14)$$P_{L}=\cos^{2}\left(\alpha +\phi_{2}/2\right)\;\cos^{2}\left(\frac{\delta -\phi_{1}}{2}\right)+\cos^{2}\left(\alpha -\phi_{2}/2\right)\;\sin^{2}\left(\frac{\delta -\phi_{1}}{2}\right)=E\left(\phi_{1},\phi_{2}\right)+O\left(\phi_{1},\phi_{2}\right)$$

Note that $P_{L}$ is a function of $\phi_{1}$, $\phi_{2}$, whereas α and δ are parameters, which identify the input SOP.

Thus, the normalised power exiting from the lower branch is given by the sum of two surfaces *O* and *E* (Figure 4a,b) consisting in the square product of trigonometric functions.

Consequently, each surface is bounded between 0 and 1, and vanishes on lines parallel to the coordinate axes, spaced by 2π.

For the first surface, *E*, the maxima are located in (cf. Equation (9a)):(15)$$\phi_{2}^{\prime }=-2\;\alpha +2m\pi \;\phi_{1}^{\prime }=\delta +2n\;\pi \;$$
and adding an odd multiple of π to either angle would cancel the first term.

Instead, for the second surface, O, the maxima are located in (cf. Equation (9b)):(16)ϕ2″=−2α+2mπϕ1″=δ+(2n+1)π

Thanks to the fact that the two sets of solutions have the optimal value for $\phi_{1}$ differing by odd multiples of π, the maxima of one surface lie on the null contour lines of the other and vice versa; this guarantees that the value of 1 will not be exceeded. The maxima position depends on the input polarisation state: the surfaces shift accordingly.

Both surfaces are shifted by the same amount and in the same direction along the $\phi_{1}$ axis upon a change in δ, whereas a variation in α produces an equal and opposite shift along $\phi_{2}$. Thus, changing δ results in a mere shift of the total surface (Figure 5b), but acting on α brings about a deformation of the surface (Figure 5c).

As shown in Equation (12), α depends on $s_{1}$ alone, so the surface shape depends on it only, i.e., is the same for all the points on “parallels” of Poincaré sphere (Section 2.2.1) with the same value of $s_{1}$, while the surfaces for different “longitudes” differ by a shift along $\phi_{1}$.

In the limit case when the $\phi_{2}$ values for the two solution sets maxima coincide, we have:(17)$$-2\;\alpha +2m\pi =2\;\alpha +2n\pi \;\Rightarrow \;\alpha =n\;\frac{\pi }{2}$$

Distinguishing between even and odd multiples of π/2, and neglecting the factor mπ (as the sign change it could introduce is cancelled by the square), we find:
(18a)$$\begin{array}{cccc} n & =2m & P_{L} & =\cos^{2}\left(\phi_{2}/2\right) \end{array}$$
(18b)$$\begin{array}{cccc} n & =2m+1 & P_{L} & =\sin^{2}\left(\phi_{2}/2\right) \end{array}$$

It is evident that the first phase shifter has no effect, as there is no dependence on $\phi_{1}$, as displayed in Figure 6.

In fact, those two solutions correspond, respectively, to
(19a)α=mπ→tanα=a2a1=0→a2=0s1=−1
(19b)$$\begin{array}{ccccc} \alpha & =\frac{\pi }{2}+m\pi & \to \;\tan\alpha & =\frac{a_{2}}{a_{1}}=\pm \infty & \to \;a_{1}=0\;s_{1}=-1 \end{array}$$
that is to say, the power flows completely either in the upper or lower coupler branch, respectively; in turn, this means that the input polarisation is one of the two principal SOPs of the 2DGC, which here are assumed by convention as horizontal or vertical.

As one of the two amplitudes is zero, its phase is not defined, thus there is no need for phase compensation.

Surfaces corresponding to input polarisations with opposite values of $s_{1}$, i.e., lying on opposite “parallels”, possess the same shape, except for a reflection around the $\phi_{1}$ axis.

In turn, this means that, for polarisations with opposite values of $s_{1}$, the values for α are complementary
(20)$$\alpha^{+}+\alpha^{-}=\pi /2$$

The intensity surfaces for orthogonal polarisations are complementary, i.e., their sum equals to 1 (Figure 7). In fact, if $\left(\phi_{1},\;\phi_{2}\right)$ are set so that all the power exits from the bottom port, then, when the orthogonal polarisation is fed into the circuit, the result is to have no power at the bottom port, in that it exits all from the top one. This fact is useful for polarisation multiplexing.

An important remark to be done is that there are always two maxima in the square with one vertex on the origin and side length of 2π in the positive axes direction (given that the thermo optic effect is exploited, it is possible to apply phase shifts of just one sign); i.e., phase shifts no greater than 2π are needed in order to recover any possible polarisation (although this does not guarantee that a drifting SOP can be tracked without interruptions, i.e., with endless operation). The blue lines connecting the different maxima in Figure 8 follow a path, which in the worst case scenario in Figure 9 has a minimum along the curve of 0.5. That means that it is possible to hop from one maxima to another suffering at most a 3 dB loss. This can be particularly useful to reduce power consumption and to avoid hitting the physical limits of the heaters. Finally, it must be highlighted that, contrarily to what Figure 5 may suggest, generally maxima and minima do not lie on lines parallel to the bisectors, as shown in Figure 8. This only occurs if α=π/4 (Figure 9).

Another important feature that is clearly apparent in Figure 8 is that for $\phi_{2}=m\pi$ the intensity is constant, independently from the value of $\phi_{1}$. In particular, Equation (14) tells that
(21)$$P_{L}\left(\phi_{1},2n\pi \right)=\cos^{2}\alpha \;P_{L}\left(\phi_{1},\left(2n+1\right)\pi \right)=\sin^{2}\alpha$$

#### 2.1.3. Use as Polarimeter

As shown in Section 2.1.1, we found the phase shifts $\phi_{1}$ and $\phi_{2}$ needed to convert the known input polarisation into the standard “TE” mode or, in other words, to inject all the power in one single output waveguide. Actually, two pairs were found for each SOP and one may wonder if it is possible to reverse the problem, i.e., to determine the SOP from the knowledge of the phase shifts that achieve optimal conversion. We show that said problem is solvable and that the existence of two solution classes generates no ambiguity, within a π shift range.

Hence, let us suppose that there exists a pair of polarisations p and $\tilde{p}$ for which there is ambiguity, that is to say, the second class of solutions of $\tilde{p}$ coincides with the first one of p or vice versa:(22)ϕ1′′=−2δ+(2nπ+1)=ϕ1˜″=−2δ˜+(2m+1)πϕ2′′=−2α+(2lπ(+1)=ϕ2˜″=−2α˜+2qπϕ1″=−2δ+(2n+1)π=ϕ1˜′′=−2δ˜+2mπϕ2″=−2α+(2lπ+1)=ϕ2˜′′=−2α˜+2qπ

In both cases, the solution is:(23)$$\tilde{\delta }=\delta +\left(2m+1\right)\pi \;\tilde{\alpha }=-\alpha +q\pi$$

If we substitute back in the input Jones vector, we find the same vector, except for a change of sign:(24)$$\tilde{A_{in}}=\left(\begin{array}{c} \cos(\tilde{\alpha })\\ \sin\left(\tilde{\alpha }\right)\;e^{i\;\tilde{\delta }} \end{array}\right)={\left(-1\right)}^{q}\left(\begin{array}{c} \cos(\alpha )\\ -\sin\left(\alpha \right)\left(-e^{i\;\delta }\right) \end{array}\right)={\left(-1\right)}^{q}\;A_{in}$$

This means that there is no ambiguity.

Once the offsets and the power needed to produce a 2π phase shift are known, the applied phase shifts $\left(\varphi_{1},\varphi_{2}\right)$ can be computed.

Next, exploiting the device periodicity, we restrict to the first period (which on the $\left(\phi_{1},\phi_{2}\right)$ plane is a square of side 2π with one vertex on the origin), and consider the reminders ${\hat{\varphi }}_{1},{\hat{\varphi }}_{2}$ modulo 2π:(25)$${\hat{\varphi }}_{1}\equiv \varphi_{1}\;\mathrm{mod}\;2\pi \;{\hat{\varphi }}_{2}\equiv \varphi_{2}\;\mathrm{mod}\;2\pi$$

It is possible to figure out to which class belongs the solution at hand; in fact, considering that 2α ranges from 0 to π, we immediately conclude that:(26)$$\begin{array}{ccccc} {\hat{\varphi }}_{2}=2\pi -2\alpha & \;\in & \left(\pi ,2\pi \right) & \;\mathrm{if} & \varphi_{2}=\phi_{2}^{\prime }=-2\alpha +2n\pi \\ {\hat{\varphi }}_{2}=\;2\alpha & \;\in & \left(0,\pi \right) & \;\mathrm{if} & \varphi_{2}=\phi_{2}^{\prime \prime }=\;2\alpha +2n\pi \end{array}$$

In the first case, the remainder of the first phase equals the longitude δ:(27)$${\hat{\varphi }}_{1}=\delta$$
while, in the second, we need to further distinguish between the lower and upper half. In fact, it is:(28)$$\phi_{1}^{\prime \prime }=\delta +\left(2m+1\right)\pi \;\to \;{\hat{\varphi }}_{1}=\left\{\begin{array}{ccc} \delta -\pi & \;\mathrm{if}\;\in & \left(0,\pi \right)\\ \delta +\pi & \;\mathrm{if}\;\in & \left(\pi ,2\pi \right) \end{array}\right.$$

The situation is shown in Figure 10.

Now that univocal behaviour has been proven, we can take advantage of Equation (21) to determine the SOP. It tells that(29)$$P_{L}\left(\phi_{1},2n\pi \right)-P_{L}\left(\phi_{1},\left(2n+1\right)\pi \right)=\cos^{2}\alpha -\sin^{2}\alpha =\cos2\alpha \;\frac{P_{L}\left(\phi_{1},\left(2n+1\right)\pi \right)}{P_{L}\left(\phi_{1},2n\pi \right)}=\tan^{2}\alpha$$
i.e., α can be extrapolated from two measurements, e.g., at $\phi_{2}=0$ and π, up to its sign
(30)$$2\alpha =\pm \arccos\left(P_{L}\left(\phi_{1},0\right)-P_{L}\left(\phi_{1},\pi \right)\right)\;\alpha \approx \pm \sqrt{\frac{P_{L}\left(\phi_{1},\pi \right)}{P_{L}\left(\phi_{1},0\right)}}$$

The first expression is more accurate when α is around 0 or π/2 ($P_{L}\left(\phi_{1},0\right)\gg P_{L}\left(\phi_{1},\pi \right)$ or vice versa) but becomes sensitive to measurement errors for α≈π/4 ($P_{L}\left(\phi_{1},0\right)\approx P_{L}\left(\phi_{1},\pi \right)$), while the second expression has a complimentary behaviour.

Next step is to set $\phi_{2}$ to this value and repeat the previous procedure on $\phi_{1}$, getting:(31)ϕ2=2αϕ1=0PL(0,2α)=cos22α+sin22αsin2(δ/2)ϕ2=2αϕ1=πPL(π,2α)=cos22α+sin22αcos2(δ/2)ϕ2=−2αϕ1=0PL(0,−2α)=cos22α+sin22αcos2(δ/2)ϕ2=2αϕ1=πPL(π,−2α)=cos22α+sin22αsin2(δ/2)

In conclusion,
(32)$$P_{L}\left(0,2\alpha \right)-P_{L}\left(\pi ,2\alpha \right)=-\sin^{2}2\alpha \;\cos\delta \;P_{L}\left(0,-2\alpha \right)-P_{L}\left(\pi ,-2\alpha \right)=\sin^{2}2\alpha \;\cos\delta$$

This allows finding δ. As with Equation (30), this formula is accurate for $P_{L}\left(0,\pm 2\alpha \right)\gg P_{L}\left(\pi ,\pm 2\alpha \right)$ or vice versa, corresponding to δ around 0 or π and 2α≈π/2; however, it is vulnerable when $P_{L}\left(0,\pm 2\alpha \right)\approx P_{L}\left(\pi ,\pm 2\alpha \right)$.

Noticing that Equation (31) can be expressed as
(33)$$\begin{array}{cc} P_{L}\left(0,2\alpha \right)=1-{\left[\sin2\alpha \;\cos(\delta /2)\right]}^{2} & \;P_{L}\left(0,-2\alpha \right)=1-{\left[\sin2\alpha \;\sin(\delta /2)\right]}^{2}\\ P_{L}\left(\pi ,2\alpha \right)=1-{\left[\sin2\alpha \;\sin(\delta /2)\right]}^{2} & \;P_{L}\left(\pi ,-2\alpha \right)=1-{\left[\sin2\alpha \;\cos(\delta /2)\right]}^{2} \end{array}$$
one can use
(34)$$\tan^{2}\left(\delta /2\right)=\frac{1-P_{L}\left(\pi ,2\alpha \right)}{1-P_{L}\left(0,2\alpha \right)}\;\mathrm{or}\;\tan^{2}\left(\delta /2\right)=\frac{1-P_{L}\left(0,-2\alpha \right)}{1-P_{L}\left(\pi ,-2\alpha \right)}$$
that are better when $P_{L}\left(0,\pm 2\alpha \right)\approx P_{L}\left(\pi ,\pm 2\alpha \right)$.

Both Equations (32) and (34) have issues for α=0 or π/2, but this is not a problem, in that δ is not well defined around those values (Figure 6).

### 2.2. Controller

If the device is used in the other direction, i.e., sending a lightwave in the lower arm and letting it exit from the 2D grating coupler, as depicted in Figure 11, then it can be used to control the polarisation state of output wave, that is to say, as a polarisation controller.

The overall transfer matrix is the adjoint of the one in Equation (2): (35)$$\begin{array}{c} \tilde{T}=M_{PS_{1}}^{†}\;M_{C_{1}}^{†}\;M_{PS_{2}}^{†}\;M_{C_{2}}^{†}={\left(M_{C_{2}}M_{PS_{2}}M_{C_{1}}M_{PS_{1}}\right)}^{†}=T^{†}=\frac{e^{i\;\psi }}{2}\;\left[\begin{array}{cc} e^{-i\;\phi_{1}}\left(1-e^{-i\;\phi_{2}}\right) & i\;e^{-i\;\phi_{1}}\left(1+e^{-i\;\phi_{2}}\right)\\ \;i\left(1+e^{-i\;\phi_{2}}\right) & \;-\;\left(1-e^{-i\;\phi_{2}}\right) \end{array}\right] \end{array}$$

The input vector will always be of the form:(36)$$A_{in}=a\left(\begin{array}{c} 0\\ 1 \end{array}\right)$$

Thus, the output polarisation is:(37)$$A_{out}=\tilde{T}A_{in}=a\;\frac{e^{i\;\psi }}{2}\left(\begin{array}{c} i\;e^{-i\;\phi_{1}}\left(1+e^{-i\;\phi_{2}}\right)\\ \;-\;\left(1-e^{-i\;\phi_{2}}\right) \end{array}\right)=\left(\begin{array}{c} a_{1}\;e^{i\;\delta_{1}}\\ a_{2}\;e^{i\;\delta_{2}} \end{array}\right)$$

Since the device is assumed to be lossless, power is conserved:(38)$$\left|\right|A_{out}{\left|\right|}^{2}=s_{0}≐a_{1}^{2}+a_{2}^{2}=a^{2}$$

The components amplitude and phase equal:(39)$$\begin{array}{cc} a_{1} & =a\sqrt{\frac{1+\cos\phi_{2}}{2}}\;\delta_{1}=\frac{\pi }{2}-\phi_{1}+\arctan\left(\frac{-\sin\phi_{2}}{1+\cos\phi_{2}}\right)\\ a_{2} & =a\sqrt{\frac{1-\cos\phi_{2}}{2}}\;\delta_{2}=\pi +\arctan\left(\frac{\sin\phi_{2}}{1-\cos\phi_{2}}\right) \end{array}$$

Consequently, the phase difference is
(40)$$\delta =\delta_{2}-\delta_{1}=\phi_{1}+\frac{\pi }{2}\left[1+\mathrm{sgn}\left(\sin\phi_{2}\right)\right]$$

The *Stokes parameters* (Equation (11)) can be explicitly derived as:(41)$$\begin{array}{cc} s_{1} & =s_{0}\;\cos\left(-\phi_{2}\right)\\ s_{2} & =s_{0}\;\sin\left(-\phi_{2}\right)\;\cos\phi_{1}\\ s_{3} & =s_{0}\;\sin\left(-\phi_{2}\right)\;\sin\phi_{1} \end{array}$$

It is convenient to take the minus sign in front of $\phi_{2}$, as the phase shifter is assumed to be placed in the lower arm. However, when compared with their usual form (1.4.2 [13]),
(42)$$\left\{\begin{array}{cc} s_{1} & =s_{0}\;\cos2\chi \;\cos2\psi \\ s_{2} & =s_{0}\;\cos2\chi \;\sin2\psi \\ s_{3} & =s_{0}\;\sin2\chi \end{array}\right.$$
it is evident (cf. Figure 12) that the axes undergo the cyclic permutation and that the new and old angles are connected by the relation (which is not the only possible solution):(43)$$\begin{array}{c} s_{1}\to s_{2}\\ s_{2}\to s_{3}\\ s_{3}\to s_{1} \end{array}\;\begin{array}{cc} 2\chi & \to \pi /2-(-\phi_{2})\\ 2\psi & \to \phi_{1} \end{array}$$

In the case at hand, it is more convenient to refer the polar and azimuthal angles not to the $s_{1}-s_{2}$ plane, as usual, but to the $s_{1}$ axis and the $s_{2}-s_{3}$ plane, respectively. The situation is depicted in Figure 12a.

Another important remark is that there are again, as it is expected thanks to reciprocity, two solution sets, which result in the same point, as in Equation (9):(44)$$\begin{array}{cccc} s_{1} & =\cos(-\varphi_{2}^{\prime }) & =\cos(-\varphi_{2}^{\prime \prime }) & =\cos2\alpha \\ s_{2} & =\sin\left(-\varphi_{2}^{\prime }\right)\;\cos\varphi_{1}^{\prime } & =\sin\left(-\varphi_{2}^{\prime \prime }\right)\;\cos\varphi_{1}^{\prime \prime } & =\sin2\alpha \;\cos\delta \\ s_{3} & =\sin\left(-\varphi_{2}^{\prime }\right)\;\sin\varphi_{1}^{\prime } & =\sin\left(-\varphi_{2}^{\prime \prime }\right)\;\sin\varphi_{1}^{\prime \prime } & =\sin2\alpha \;\sin\delta \end{array}$$

In practice, when a given SOP corresponding to a point on Poincaré sphere is to be generated, the phases to be applied are, considering both solution sets:(45)$$\phi_{1}=\arctan\left(\frac{s_{3}}{s_{2}}\right)-m\pi \;\phi_{2}=-{\left(-1\right)}^{m}\arccos\left(\frac{s_{1}}{s_{0}}\right)+2n\pi$$
(to be compared with Equation (9)).

#### 2.2.1. Properties of the Poincaré Sphere

At this point, it must be recalled that the two forms of Stokes parameters in Equations (41) and (42) refer to the observer (on our case, the 2DGC) and the polarisation ellipse frames, respectively.

The flexibility offered by this method of representation provides a physical and insightful point of view particularly adapted to solve graphically otherwise cumbersome algebraic systems. In addition, let us point out that, as in [14,15], this representation using a Bloch sphere can be conveniently used for a quantum treatment of the device as there is a direct correspondence from Stokes parameters to the density operator. In the observer frame, the polarisation ellipse with semi axes *a* and *b* is tilted by an angle ψ with respect to the *x* axis and is inscribed inside a rectangle of sides $a_{1}$ and $a_{2}$ (Figure 13). The vertical component of the Jones vector $A_{out}$ is phase shifted by an angle δ (Equation (40)).

The angles α and χ are defined as:(46)$$\tan\alpha =\frac{a_{2}}{a_{1}}\;\tan\chi =\mp \frac{b}{a}$$

They are connected to the aspect ratio of the black and green rectangles in Figure 13. The angle χ is usually called ellipticity and α is termed auxiliary angle.

The Jones vector in the observer frame can be expressed as
(47)$$A=\left(\begin{array}{c} \cos\alpha \\ \sin\alpha \;e^{i\delta } \end{array}\right)$$
with respect to the basis given by the horizontal and vertical linear polarisations (TE and TM or H and V) usually used in quantum optics.

The following relations (1.4.2 [13]) hold between the angles pairs in the two coordinates systems in Figure 12:(48)$$\left\{\begin{array}{cc} \tan2\psi & =\tan2\alpha \;\cos\delta \\ \sin2\chi & =\sin2\alpha \;\sin\delta \end{array}\right.$$

About Poincaré sphere, most textbooks make clear that “parallels” and “meridians” with respect to the $s_{3}$ axis, i.e., points with the same value of χ or ψ, correspond to SOPs with the same ellipticity or tilt angle with respect to the observer’s *x* axis, respectively.

If instead the other coordinates are considered, then points with the same polar angle 2α referred to $s_{1}$ correspond to ellipses inscribed in the same rectangle with sides $a_{1}$ and $a_{2}$. Instead, points with the same “longitude” δ are associated to polarisation states with the same phase shift value between their two components, as seen in the xOy frame.

Notice that the share of power of a SOP given by Equation (47) that is let pass by an analyser whose principal state is, for instance, *H*, is
(49)$$P_{A-H}={\left(A\;\cdot \;H\right)}^{2}=\cos^{2}\alpha$$
that is basically Malus’ law in terms of Jones vectors. Considering that the dot product between the corresponding Stokes vectors (excluding $s_{0})$ is ${\vec{s}}_{A}\;\cdot \;{\vec{s}}_{H}=\cos2\alpha$, the same relation can be written
(50)$$P_{A-H}=\frac{{\vec{s}}_{A}\;\cdot \;{\vec{s}}_{H}+1}{2}$$

The validity of this formula is not limited to the particular SOP basis considered in the derivation, since a change of basis simply rotates the whole sphere. This fact tells us that the SOPs whose points on the sphere are apart by the same angle have the same power coupling.

#### 2.2.2. Generalised Poincar é/Bloch Sphere

For a generic photonic circuit made up of two waveguides, one can associate [16] to the pair of complex amplitudes in the two guides the Stokes parameters as defined in Equation (41).

Therefore, one can associate a point on the Poincaré sphere (also called Bloch sphere when generalised) with a given quadruple corresponding to the given couple of complex amplitudes. As the complex amplitudes vary upon propagation along the circuit, the corresponding point undergoes a rotation (which is the product of the rotations brought about by the several circuit stages).

Note that, in this formulation, a point on the generalised sphere does not correspond to a polarisation state, in that the two fields are generally located in distinct waveguides, whereas for a plane wave (or a field pattern in free space) the two orthogonal field components are in the same place and do overlap.

However, one can reconnect to the polarisation state in this way: for any section of the circuit, to the pair of complex amplitudes in the two waveguides corresponds a certain point on the sphere. If the circuit, in the considered section, were connected to a 2D grating coupler, then the complex amplitudes of *TE* and *TM* components would equal (except for the losses) the guided ones, so the point corresponding to the polarisation state of output light would coincide with the one corresponding to the couple of guided fields.

The convenience of this approach lies in the possibility to have a visual representation of each circuit component effect, as it results in the rotation by a certain angle and around a given axis of the entire sphere.

As shown in (p. 67 [12,16]), the effect of a phase shifter is a rotation around the $s_{1}$ axis by the differential angle Δϕ (however, the sign depends on the adopted convention) and a synchronous coupler provokes a rotation around the $s_{2}$ axis by the double of the amplitude coupling κ. For an asynchronous coupler, the rotation axis lies in the $s_{1}-s_{2}$ plane and the rotation angle is given by the same rule as for a synchronous coupler.

In our case, the MMIs produce a π/2 rotation around $s_{2}$.

The states with $s_{1}\pm 1$ correspond to all the power in the top and bottom waveguide, respectively, and are labelled as $E_{1}$ and $E_{2}$ on the figures.

#### 2.2.3. Device Operation

The effect of our circuit is, starting from the point labelled as $E_{2}$, to rotate clockwise around $s_{2}$ by a right angle, then clockwise (because the second phase shifter in placed on the lower branch) about $s_{1}$ by an angle $\phi_{2}$, again clockwise by a right angle around $s_{2}$, and eventually counter clockwise by $\phi_{1}$ around $s_{1}$, respectively.

The path is travelled backwards in the compensator operation.

This is shown in Figure 14a,b, for the two solution sets.

The starting point $E_{2}$ corresponds to the Jones vector in Equation (36), a “vertical” linear polarisation. This point is brought into $E_{L}$ by the first coupler, then in $P^{\prime \prime }=\left(\pi /2,-\pi /2\mp 2\alpha \right)$ by phase shifter $\phi_{2}$, in $P^{\prime }=\left(2\alpha ,0/\pi \right)$ (a “linear” SOP, as $s_{3}=0$, represented in the figures with a yellow dashed circle) by the other coupler and finally in P=(2α,δ) by $\phi_{1}$.

Another way of explaining the device operation is to consider it as a Mach–Zehnder Interferometer (MZI) followed by the phase shifter $\phi_{1}$. As shown in Appendix B, a MZI behaves as a (non-endless) half-wave plate [10], thus it causes a rotation of π about an axis lying on the $s_{1}-s_{2}$ equatorial plane, with azimuth
(51)$$\Theta =-\phi_{2}/2+\pi /2=\pi /2\pm \alpha -m\pi$$
with respect to the $s_{1}$ axis (the minus sign in front of $\phi_{2}$ is due to the fact that it is applied to the lower waveguide).

Thus, shifter 1 has the role of carrying the SOP from the point $P^{\prime }$ on the “equator”, to its destination point P, as displayed in Figure 14a,b.

Even if the previous description allows a better understanding the device operation, it is of great interest to determine the overall rotation. The following derivation is based on (2.6.2 in [12]). To comply with its notation for phase shifter matrices (Table 2.1 in [12]), it is better to reformulate Equation (35) as (note that the phase shifts are opposite in sign, for the phase shifters are placed on different branches):(52)T˜=isin(ϕ2/2)e−iϕ1/2−icos(ϕ2/2)e−iϕ1/2icos(ϕ2/2)eiϕ1/2−isin(ϕ2/2)eiϕ1/2==cosϕ22−π2e−i(ϕ1/2−π/2)sinϕ22−π2e−i(ϕ1/2+π/2)−sinϕ22−π2ei(ϕ1/2+π/2)cosϕ22−π2ei(ϕ1/2−π/2)

Now, to get the rotation matrix for Stokes’ parameters, the above formula must be put in form compliant with the general one for unitary matrices given in (p. 51 in [12]):(53)$$U=\left[\begin{array}{cc} e^{iℵ}\;\cos\kappa & -e^{i\beta }\;\sin\kappa \\ e^{-i\beta }\;\sin\kappa & e^{-iℵ}\;\cos\kappa \end{array}\right]$$

Comparing the previous two equations, it is clear that:(54)$$\kappa =-\frac{\phi_{2}-\pi }{2}\;ℵ=-\frac{\phi_{1}-\pi }{2}\;\beta =-\frac{\phi_{1}+\pi }{2}$$

In the reciprocal case where the device works as a compensator, the sign of κ and *ℵ* changes, while β retains its sign thanks to transposition.

To determine the global rotation, we insert the parameters of Equation (54) into the general rotation matrix given in (p. 67 [12]), getting:(55)R=−cosϕ2cosϕ1sinϕ20cosϕ1sinϕ2cosϕ1cosϕ2−sinϕ1sinϕ1sinϕ2sinϕ1cosϕ2−cosϕ1

For the compensator operation, the corresponding rotation matrix is the inverse of the one above, namely its transpose.

Substituting the values for the two solution sets:(56)$$\begin{array}{cc} R^{\prime /\prime \prime }\left(\alpha ,\delta \right) & =\left[\begin{array}{ccc} -\cos(2\alpha ) & \mp \sin(2\alpha ) & 0\\ -\cos\delta \;\sin(2\alpha ) & \pm \cos\delta \;\cos(2\alpha ) & \pm \sin\delta \\ -\sin\delta \;\sin(2\alpha ) & \pm \sin\delta \;\cos(2\alpha ) & \mp \cos\delta \end{array}\right]\\ R^{\prime /\prime \prime }\left(\vec{s}\;\right) & =\left[\begin{array}{ccc} -s_{1} & \;\mp \sqrt{s_{2}^{2}+s_{3}^{2}} & \;0\\ -s_{2} & \;\pm \frac{s_{1}\;s_{2}}{\sqrt{s_{2}^{2}+s_{3}^{2}}} & \;\pm \frac{s_{3}}{\sqrt{s_{2}^{2}+s_{3}^{2}}}\\ -s_{3} & \;\pm \frac{s_{1}\;s_{3}}{\sqrt{s_{2}^{2}+s_{3}^{2}}} & \;\mp \frac{s_{2}}{\sqrt{s_{2}^{2}+s_{3}^{2}}} \end{array}\right] \end{array}$$

One can easily check that $R^{\prime /\prime \prime }\left(\alpha ,\delta \right)$ brings $E_{2}$ into *P*, as it should:(57)$$R^{\prime /\prime \prime }\left(\alpha ,\delta \right)\left(\begin{array}{c} -1\\ 0\\ 0 \end{array}\right)=\left(\begin{array}{c} \cos(2\alpha )\\ \sin(2\alpha )\;\cos\delta \\ \sin(2\alpha )\;\sin\delta \end{array}\right)=\left(\begin{array}{c} s_{1}\\ s_{2}\\ s_{3} \end{array}\right)=P$$

The rotation axis is (Section 9.3.1 in [17]): (58)$$\vec{\Omega }=\left(\begin{array}{c} R_{32}-R_{23}\\ R_{13}-R_{31}\\ R_{21}-R_{12} \end{array}\right)=A\left(\begin{array}{c} \tan(\phi_{2}/2)\\ 1\\ \tan(\phi_{1}/2) \end{array}\right)$$
the multiplicative factor *A* is connected to the vector norm, which, however, is irrelevant. The two solution classes have different rotation axes (Figure 15): (59)$${\vec{\Omega }}^{\prime }=A^{\prime }\left(\begin{array}{c} -\tan(\alpha )\\ 1\\ \tan(\delta /2) \end{array}\right)\;{\vec{\Omega }}^{\prime \prime }=A^{\prime \prime }\left(\begin{array}{c} \tan(\alpha )\\ 1\\ -\cot(\delta /2) \end{array}\right)$$

In general, they are not perpendicular to each other: (60)$${\vec{\Omega }}^{\prime }\;\cdot \;{\vec{\Omega }}^{\prime \prime }\propto -\tan^{2}\alpha$$

However, each rotation axis is perpendicular to the other axis’s projection on the $s_{2},s_{3}$ plane: (61)$${\vec{\Omega }}^{\prime }\;\cdot \;{\vec{\Omega }}_{s_{2}s_{3}}^{\prime \prime }=A^{\prime }\left(\begin{array}{c} -\tan(\alpha )\\ 1\\ \tan(\delta /2) \end{array}\right)\;\cdot \;A^{\prime \prime }\left(\begin{array}{c} 0\\ 1\\ -\cot(\delta /2) \end{array}\right)=0$$

The rotation angle is given by (Section 9.3.1 in [17]):(62)$$\cos\left(\Gamma \right)=\frac{\mathrm{tr}(R)-1}{2}=2{\left(\sin\left(\phi_{1}/2\right)\;\sin\left(\phi_{2}/2\right)\right)}^{2}-1$$

Using a trigonometric identity, the formula reduces to:(63)$$\cos\left(\Gamma /2\right)=\pm \sin\left(\phi_{1}/2\right)\;\sin\left(\phi_{2}/2\right)$$

For the two solutions sets, it reads
(64)$$\begin{array}{cc} \cos(\Gamma^{\prime }/2) & =\mp \sin(\delta /2)\sin(\alpha )\\ \cos(\Gamma^{\prime \prime }/2) & =\pm \cos(\delta /2)\sin(\alpha ) \end{array}$$

The overall circuit effect is shown in Figure 16.

## 3. Frequency Response

To deduce the light frequency dependence of the device at hand, the first order expansion for the waveguides effective index
(65)$$n_{eff}\left(\lambda \right)=n_{eff_{0}}\;\frac{\lambda }{\lambda_{0}}-n_{g_{0}}\;\frac{\lambda -\lambda_{0}}{\lambda_{0}}$$
is inserted in the formula for the phase shift:(66)$$\phi_{i}=\frac{2\pi }{\lambda }\;\Delta_{tb}\left(n\;L_{i}\right)=\frac{2\pi \;\Delta_{tb}\left(n_{eff_{0}}\right)\;L_{i}}{\lambda_{0}}\left[1-\frac{\Delta_{tb}\;n_{g_{0}}}{\Delta_{tb}\;n_{eff_{0}}}\left(1-\frac{\lambda_{0}}{\lambda }\right)\right]$$

The different effective indexes of the top and bottom arms of a phase shifter result from their different temperature, since thermo optic effect is exploited in silicon. In Appendix A, it is shown that, at first order, the group index too is proportional to the temperature shift. Then,
(67)$$\frac{\Delta_{tb}\;n_{g_{0}}}{\Delta_{tb}\;n_{eff_{0}}}=\frac{\alpha_{g}\;\Delta_{tb}\left(T-T_{0}\right)}{\alpha_{eff}\Delta_{tb}\left(T-T_{0}\right)}=\frac{\alpha_{g}}{\alpha_{eff}}≐R_{ge}$$

In conclusion, the phase shift has the dependence
(68)$$\phi_{i}=\phi_{i_{0}}\left(1-R_{ge}\frac{\Delta \lambda }{\lambda }\right)=\phi_{i_{0}}\left(1+R_{ge}\frac{\Delta f}{f_{0}}\right)$$

In the $\left(\phi_{1},\phi_{2}\right)$ plane, once a given solution at $f_{0}$ has been selected, the point $\left(\phi_{1}\left(f\right),\phi_{2}\left(f\right)\right)$ lies on a line passing through $\left(\phi_{1_{0}},\phi_{2_{0}}\right)$ and the origin (Figure 17): (69)$$\vec{\phi }\left(f\right)=\left(\begin{array}{c} \phi_{1}\left(f\right)\\ \phi_{2}\left(f\right) \end{array}\right)=\left(\begin{array}{c} \phi_{1_{0}}\\ \phi_{2_{0}} \end{array}\right)\left(1+R_{ge}\;\frac{\Delta f}{f_{0}}\right)=\vec{\phi_{0}}\left(1+R_{ge}\;\frac{\Delta f}{f_{0}}\right)$$

The “phase speed” or sensitivity to a frequency shift df
(70)$$\overset{•}{\phi_{i}}=\frac{d\phi_{i}}{df}=\phi_{i_{0}}R_{ge}\frac{1}{f_{0}}$$
is proportional to the phase shift $\phi_{i_{0}}$ applied at the central frequency $f_{0}$, hence choosing higher order solutions, which are farther from the origin, entails a larger sensitivity to frequency, that is, a narrower bandwidth.

### 3.1. Compensator

The several factors appearing in the expression for the intensity surface $P_{L}$, Equation (14), are function of the difference
(71)$$\phi_{i}-\phi_{i_{0}}=\phi_{i_{0}}R_{ge}\frac{\Delta f}{f_{0}}$$

To find the Free Spectral Range (FSR), i.e., the frequency spacing—if any—between two consecutive peaks of the intensity surface, the above difference should be an integer multiple of 2π, for both phase shifters. Unfortunately, since the two central frequency phase shifts are in general different, there are two different FSRs (if it makes any sense):(72)$$\Delta f_{i}=\frac{f_{0}}{R_{ge}}\frac{2m_{i}\pi }{\phi_{i_{0}}}$$

It is possible to define an FSR just in the particular case when $\phi_{1_{0}}$ and $\phi_{2_{0}}$ are commensurable
(73)$$\frac{\phi_{2_{0}}}{\phi_{1_{0}}}=\frac{p}{q}\Rightarrow \;FSR=\frac{f_{0}}{R_{ge}}\frac{2p\pi }{\phi_{2_{0}}}=\frac{f_{0}}{R_{ge}}\frac{2q\pi }{\phi_{1_{0}}}$$
where *p* and *q* are relatively prime integers.

#### 3.1.1. Contour Curves

In the following (Section 3.1.2), we consider contour curves, in particular the one for which $P_{L}\left(\phi_{1},\phi_{2}\right)=1/2$.

Let us consider the general case of a contour curve for the level *b*:(74)$$\cos^{2}\left(\phi_{2}/2+\alpha \right)\;\cos^{2}\left(\frac{\phi_{1}-\delta }{2}\right)+\cos^{2}\left(\phi_{2}/2-\alpha \right)\;\sin^{2}\left(\frac{\phi_{1}-\delta }{2}\right)=b$$

After applying some trigonometric identities, we arrive to an expression for $\phi_{1}$ as a function of $\phi_{2}$:(75)$$\cos\left(\phi_{1}-\delta \right)=\frac{1-2b}{\sin\phi_{2}\;\cdot \;\sin2\alpha }+\cot\phi_{2}\;\cdot \;\cot2\alpha$$

#### 3.1.2. Bandwidth

The −3 dB bandwidth is given by the equation
(76)$$P_{L}\left(\phi_{1}\left(f\right),\phi_{2}\left(f\right)\right)=1/2$$

In Section 3.1.1, an expression for the contour curves of level *b* was found; in this particular case of b=1/2, one term vanishes and the formula becomes:(77)$$\cos\left(\phi_{1}\left(f\right)-\delta \right)=\cot\phi_{2}\left(f\right)\;\cdot \;\cot2\alpha$$

Depending on the solution set
(78)$$\cos\left(\phi_{1}\left(f\right)-\delta \right)=\pm \cos\left(\phi_{1_{0}}^{{}^{\prime }{/}^{\prime \prime }}R_{ge}\;\frac{\Delta f}{f_{0}}\right)\;\cot\phi_{2_{0}}^{{}^{\prime }{/}^{\prime \prime }}=\mp \cot2\alpha$$

The bandwidth $\Delta f_{3\mathrm{dB}}$ is given by the implicit equation
(79)$$\cos\left(\phi_{1_{0}}^{j}R_{ge}\;\frac{\Delta f}{f_{0}}\right)=-\cot\phi_{2_{0}}^{j}\;\cdot \;\cot\left[\phi_{2_{0}}^{j}\left(1+R_{ge}\;\frac{\Delta f}{f_{0}}\right)\right]$$
where *j* indicates the solution class.

Graphically, the equation corresponds to finding the frequency shifts for which the lines corresponding to the phase pair intersect the contour curves for the level 1/2, which lie closest to the considered solution, as in Figure 17.

The bandwidth can be visualised as the distance between two such intersections, divided by the solution distance from the origin (because the “phase speed” is proportional to it, see Equation (70)) and multiplied by $f_{0}$.

#### 3.1.3. Spectrum

The spectrum depends on the input SOP and is given by the expression for $P_{L}$ when the frequency dependence as in Equation (68) is included
(80)$$P_{L}\left(f\right)=\cos^{2}\left(\phi_{2}/2+\alpha \right)\;\cos^{2}\left(\frac{\phi_{1}-\delta }{2}\right)+\cos^{2}\left(\phi_{2}/2-\alpha \right)\;\sin^{2}\left(\frac{\phi_{1}-\delta }{2}\right)$$

For a better insight, the spectrum is found from the intersection with the intensity surface of the plane perpendicular to the $\left(\phi_{1},\phi_{2}\right)$ plane and passing by the line traced by $\vec{\phi }\left(f\right)$ as in Equation (69). Figure 18 displays the spectra for the same situation as in Figure 17. The relation between spectrum shapes and the corresponding intensity surface for the considered SOP would be more apparent if a linear, rather than logarithmic, vertical scale had been plotted against frequency, instead of wavelength.

#### 3.1.4. Effect of Unbalance

Besides the temperature difference between the two phase shifters in a given stage, differences in the applied phase shifts may arise because of differences in any of the following waveguide properties:lengthwidththicknessmaterial composition (e.g., doping)

All of those effects are permanent, that is to say, are present even if no external control signal is applied to the circuit, thus resulting in an offset, which must be compensated, in order to apply the correct phase shift.

From the modelling perspective, just length asymmetry appears explicitly, while the other three parameters give rise to variations in quantities such as $n_{eff_{0}}$, $n_{g_{0}}$, $\alpha_{eff}$ and $\alpha_{g}$. For the sake of simplicity, the last two contributions have been neglected.

Taking the top branch as the unbalanced one
(81)$$\begin{array}{ccccc} L_{b} & =L & & L_{t} & =L_{b}+\Delta l=L+\Delta l\\ n_{eff_{0_{b}}} & =n_{eff_{0}} & & n_{eff_{0_{t}}} & =n_{eff_{0}}+\Delta n_{eff_{0}}\\ n_{g_{0_{b}}} & =n_{g_{0}} & & n_{g_{0_{t}}} & =n_{g_{0}}+\Delta n_{g_{0}} \end{array}$$
the phase shift from Equation (66) becomes
(82)$$\phi =\phi_{0}+p\;\Delta f/f_{0}$$
where
(83)$$\begin{array}{cc} \phi_{0} & =\frac{2\pi }{\lambda_{0}}\left\{n_{eff_{0}}\Delta l+\Delta n_{eff_{0}}L+\alpha_{eff}\left[\left(T_{t}-T_{b}\right)\;L+\left(T_{t}-T_{0}\right)\Delta l\right]\right\}\\ p & =\frac{2\pi }{\lambda_{0}}\left\{n_{g_{0}}\Delta l+\Delta n_{g_{0}}L+\alpha_{g}\left[\left(T_{t}-T_{b}\right)\;L+\left(T_{t}-T_{0}\right)\Delta l\right]\right\} \end{array}$$
while, in the balanced case, both $\phi_{0}$ and *p* are proportional to ΔT (thus, one could solve for ΔT from $\phi_{0}$ and then substitute to find the slope *p*), now there are two unknown variables: the two shifters’ temperatures.

Nevertheless, this is no problem, in that we can consider $T_{b}$ as a free parameter and express $T_{t}$ as a function of it:(84)$$T_{t}=\frac{\phi_{0}\;\frac{\lambda_{0}}{2\pi }-\left(n_{eff_{0}}\Delta l+\Delta n_{eff_{0}}L\right)+\alpha_{eff}\left(T_{b}L+T_{0}\Delta l\right)}{\alpha_{eff}\left(L+\Delta l\right)+\Delta \alpha_{eff}L}$$

Thus, the slope is:(85)$$\begin{array}{cc} p=\phi_{0}\;R_{ge}+\Lambda_{ge}\;\Delta l+\Gamma_{ge}L & =s_{0}+\Delta s\\ \Lambda_{ge}≐\frac{2\pi }{\lambda_{0}}\left[n_{g_{0}}-\frac{\alpha_{g}}{\alpha_{eff}}n_{eff_{0}}\right]\;\Gamma_{ge} & ≐\frac{2\pi }{\lambda_{0}}\left[\Delta n_{g_{0}}-\frac{\alpha_{g}}{\alpha_{eff}}\Delta n_{eff_{0}}\right] \end{array}$$

The first factor appearing in the formula for *p* is the one in the ideal case or $p_{0}$, while the other two come from asymmetry, and can be gathered in a Δp term.

Considering the relation between the two phases,
(86)$$\phi_{2}=\phi_{2_{0}}+p_{2}\;\frac{\Delta f}{f_{0}}=\phi_{2_{0}}+\frac{p_{2}}{p_{1}}\left(\phi_{1}-\phi_{1_{0}}\right)$$
we see that the phase pair still traces a line in the $\left(\phi_{1},\phi_{2}\right)$ plane as a result of frequency shifts, but in general this line does not pass by the origin, as in general the ratio in front of $\phi_{1}-\phi_{1_{0}}$ differs from $\phi_{2_{0}}/\phi_{1_{0}}$.

#### 3.1.5. Bandwidth Variation

To find the bandwidth in this non-ideal case, the expression for the phase in Equation (82) is replaced in Equation (79)
(87)$$\cos\left(p_{1}\frac{\Delta f}{f_{0}}\right)=\mp \cot2\alpha \;\cdot \;\cot\left(\phi_{2_{0}}^{j}+p_{2}\frac{\Delta f}{f_{0}}\right)$$

As a shorthand notation, the term $B_{id}$ is used in place of $\Delta f/f_{0}$ in the ideal, balanced case, whereas ΔB stands for the variation of $B_{id}$.

In the real case, Equation (79) can be rewritten as:(88)$$\begin{array}{cc} \cos & \left[\left(p_{1_{0}}+\Delta p_{1}\right)\left(B_{id}+\Delta B\right)\right]=\\ & =-\cot\phi_{2_{0}}^{j}\;\cdot \;\cot\left[\phi_{2_{0}}^{j}+\left(p_{2_{0}}+\Delta p_{2}\right)\left(B_{id}+\Delta B\right)\right] \end{array}$$

Neglecting higher order terms, one gets
(89)$$\Delta B\approx -\frac{\sin\left(p_{1_{0}}B_{id}\right)\Delta p_{1}+\cot\phi_{2_{0}}^{j}\csc^{2}\left(\phi_{2_{0}}^{j}+p_{2_{0}}B_{id}\right)\Delta p_{2}}{\sin\left(p_{1_{0}}B_{id}\right)p_{1_{0}}+\cot\phi_{2_{0}}^{j}\csc^{2}\left(\phi_{2_{0}}^{j}+p_{2_{0}}B_{id}\right)p_{2_{0}}}\;B_{id}$$

It tells us that the effect of imbalance is not necessarily detrimental, i.e., to reduce the bandwidth, provided that the ratio in front of $B_{id}$ is negative. However, since Equation (89) is highly dependent on the input SOP, there is no easy trend that can be estimated and the only conclusion is that the bandwidth is highly dependent on the input.

### 3.2. Controller

Putting the frequency dependence of the phase shifts given by Equation (68) into the formula for the Stokes parameters of the SOP coming out of the circuit, we arrive to
(90)s1=s0cosϕ101+RgeΔff0cos−ϕ201+RgeΔff0s2=s0cosϕ101+RgeΔff0sin−ϕ201+RgeΔff0s3=s0sinϕ101+RgeΔff0sin−ϕ201+RgeΔff0

Given that the two phases $\phi_{1}$ and $\phi_{2}$ are proportional to each other, the motion of the SOP on Poincaré sphere can be described by a single variable Θ, for instance taken equal to $\phi_{1}$, so that the previous equations become:(91)s1=s0cos(θ)cos(mθ)s2=s0cos(θ)sin(mθ)s3=s0sin(θ)sin(mθ)
and the ratio
(92)$$m=-\phi_{2_{0}}/\phi_{1_{0}}$$
acts as a parameter. This family of curves is named Clélie and some examples are shown in Figure 19a for several values of *m*. Its projection on the $s_{2}-s_{3}$ plane is a plane curve called rhodonea or rose, with polar equation
(93)ρ=|sin(mθ)|

Polarisations whose Stokes’ parameters are such that:(94)2α/δ=−m
lie on the same Clélie; the result of a frequency shift is to move the point on such curve.

Nonetheless, because of the existence of a countable infinity of phase shift pairs, which yield the same SOP, *m* takes on different values for the same polarisation, which means that there are many curves passing through a given point on the sphere.

Then, depending on the chosen solution, the point would follow a different path, on the corresponding curve.

If the curve is expressed as in Equation (91), then problems would arise for $\phi_{1_{0}}=0$ (linear polarisations) or for values close to it, as *m* would diverge. This issue can be removed, however; in fact, it suffices to redefine said equation as
(95)s1=s0cos(ϑ/m)cos(ϑ)s2=s0cos(ϑ/m)sin(ϑ)s3=s0sin(ϑ/m)sin(ϑ)
with $\vartheta =-\phi_{2}$. This second situation is shown in Figure 19b.

#### 3.2.1. Bandwidth

For a given central frequency SOP with Stokes parameters ${\left(s_{1},s_{2},s_{3}\right)}_{0}$, the 3 dB optical bandwidth can be visualised as the frequency excursion Δf necessary to reach a perpendicular (Section 2.2.1) Stokes vector ${\vec{s}}_{3\mathrm{dB}}$ (from Equation (41)) lying on the same Clélie: (96)$${\vec{s}}_{3\mathrm{dB}}\;\cdot \;{\vec{s}}_{0}=0$$

When expanded, the above relation becomes
(97)$$\cos2\alpha \;\cos\phi_{2}-\sin2\alpha \;\sin\phi_{2}\;\cos\left(\phi_{1}-\delta \right)=0$$
that is essentially Equation (79). Using Equation (50), the above procedure can be generalised to find the contour lines as in Section 3.1.1.

#### 3.2.2. Effect of Unbalance

In the case of unbalanced shifters, as shown in Section 3.1.4, the ratio between the two phases is no longer independent from frequency, thus the above treatment is no longer valid. However, there is still a linear (or rather affine) relationship between the two phases (Equation 86)
(98)$$\phi_{2}=\phi_{2_{0}}+\frac{s_{2}}{s_{1}}\left(\phi_{1}-\phi_{1_{0}}\right)=-r\phi_{1}+\left(\phi_{2_{0}}+r\phi_{1_{0}}\right)=-r\phi_{1}-\gamma_{0}$$
where *r* is the ratio between the slopes of the two phases and in general differs from $m=-\phi_{2_{0}}/\phi_{1_{0}}$, since
(99)$$r≐-\frac{s_{2}}{s_{1}}=-\frac{s_{2_{0}}+\Delta s_{2}}{s_{1_{0}}+\Delta s_{1}}\neq -\frac{s_{2_{0}}}{s_{1_{0}}}=-\frac{\phi_{2_{0}}}{\phi_{1_{0}}}=m$$

The evolution with frequency of the SOP can again be described with a single parameter.
(100)s1=s0cos(θ)cos(rθ+γ0)s2=s0cos(θ)sin(rθ+γ0)s3=s0sin(θ)sin(rθ+γ0)

The curve remains a Clélie, just rotated by an angle $\gamma_{0}$ around the $s_{1}$ axis and with a different parameter *r* instead of *m*. While previously phase pairs lying on the same line passing by the origin of the $\left(\phi_{1},\phi_{2}\right)$ plane corresponded to points on the same Clélie on Poincaré sphere, now the same is true for points situated on the line given by Equation (99).

## 4. Characterisation

The predictions of the above treatment were tested using the setup in Figure 20: the SOP of the laser beam was matched to the SPGC one with a fibre polarisation controller (FPC). The light from the device under test (DUT) is sent to a −10 dB fibre beam splitter. A tenth of the power went to the power meter, to capture the spectrum, while the rest was sent to the polarimeter. Another FPC was placed in front of the polarimeter for calibration purposes.

The DUT was used in the controller configuration only because it was faster to measure many SOPs, as it was not necessary to manually act on the first FPC, which would involve several trials and errors before reaching the desired SOP to feed to the DUT.

The DUT was a test structure from the second version of the miniROADM [7], substantially with the same properties as that in [5], except for a balanced arrangement, with dummy heaters on the unused arms for a broader optical bandwidth, limited just by the GCs. Moreover, the top output was also connected to a GC, while in [5] it was terminated by a monitor photodiode. Finally, −20 dB waveguide taps were coupled to each branch, for easier characterisation.

Both versions were fabricated by commercial CMOS foundries, IMEC for the one of [5,7] and CMC-IME for the second one, respectively.

The hypothesis described in Section 1.1 was checked, as a preliminary stage, on a test structure consisting in a 2DGC whose outputs were both terminated by a SPGC. It was found that the SOPs that maximise the output power at either port were orthogonal to each other and that the transmission spectra corresponding to those SOPs overlapped. This confirmed that the 2DGC did not exhibit any significant Polarisation Dependent Loss (PDL).

The calibration was performed to have the polarimeter to display the actual SOP on the 2DGC, resulting in a reading consistent with Figure 12b and Figure 19. Indeed, even with a correctly calibrated instrument, it would display a different SOP from that on the 2DGC, because of the offset introduced by the optical fibres in between and the setup in general. The calibration purpose was to remove said offset.

The effect of a birefringent element is a sphere rotation. To uniquely determine it, two pairs of points (before and after the transformation) are needed. For the first pair, we used the SOP that maximises the power coupled to a SPGC, and the second FPC was adjusted so that the polarimeter displayed a linear horizontal SOP (H, $s_{1}=1$). This step was performed on a waveguide clip, terminated with SPGCs at both ends, a test structure usually included in evaluation chips to measure GC insertion loss (IL) and waveguide propagation losses. The SOP that maximised the power from the top output of the 2DGC was then made to correspond to a linear polarisation inclined by 45° (D, $s_{2}=1$). Such a calibration should be repeated to compensate the SOP drift due to fibres, which likely caused the visible difference between measured and estimated points on the parallel with α=30° in Figure 21.

In the first set of measurements, the Stokes parameters were read on the polarimeter and compared with the values obtained by putting into Equation (41) the dissipated power on each heater, as measured on the source meter. Given that the phase shift $\phi_{n}$ was produced by leveraging silicon thermo optic effect, it was assumed to be proportional to the power $P_{n}$ dissipated by the heater *n*:(101)$$\phi_{n}=2\pi \;P_{n}/P_{2\pi }$$
where $P_{2\pi }$ stands for the thermal efficiency, namely the power required to obtain a 2π phase shift. For data points where just one heater was active ($s_{2}=0$ and $s_{3}=$ in Figure 21), this assumption was found to be valid. The above expression was used in Equation (41) to get the Stokes parameters predicted by our model.

The dissipated power and the voltage drop displayed by the power supply does not exactly correspond to that on the heater, especially when both heaters are active, because of the parasitic resistance $R_{G}$ of the common ground electrode. To model this effect, we considered a star circuit, with $R_{G}$ at the bottom. Changing the voltage applied to one heater while keeping the other constant results in a variation of the voltage drop on $R_{G}$, thus of the current and power on the heater, that should remain unaltered. Thus, $R_{G}$ caused an electrical crosstalk between the two phase shifters. In the usual case where the voltages $V_{1}$ and $V_{2}$ applied to heater (resistor) $R_{1}$ and $R_{2}$, respectively, are both positive, increasing $V_{2}$ while keeping $V_{1}$ constant results in a reduction of $P_{1}$. Applying a current instead of a voltage bias should avoid this issue, because the power supply is in series to the heater. Moreover, it was found that the resistance increased linearly with the dissipated power:(102)$$R=R_{0}+\gamma P$$

We attributed this behaviour to the fact that in a metal the resistivity is proportional to the temperature. The main consequence of this non-linearity was a smaller dissipated power for the same applied voltage.

The thermal efficiency was of 76 ± 1 mW/cycle. The result is shown in Figure 21.

Another useful measurement was to sweep over one of the two phase shifts while keeping the other constant.

As shown in Figure 22, the point did not exactly trace parallels and meridians on Poincaré sphere. The curves did not close on themselves after a 2π shift because of the already described electrical crosstalk due to the parasitic resistance of the common ground electrode. With reference to Figure 22a, increasing $P_{1}$ to scan in “longitude” decreased $P_{2}$, hence the polar angle, thus, the point travelled on spirals instead of parallels.

When the inverse operation was performed, as in Figure 22b, a raise of the polar angle provoked a longitude diminution, which caused what would have been a meridian to precess, such as the curve of a satellite in polar orbit.

## 5. Conclusions

The operation of the silicon photonic circuit proposed in [6] has been analysed for both polarisation compensator and controller configurations. The behaviour at the central wavelength as well as the frequency dependence have both been considered. The whole treatment has been derived using a geometrical representation based on phasors and the Poincaré sphere. It has been shown that, thanks to this representation, the functioning of the device can be intuitively understood and analysed. This analysis has shown that any input SOP can be compensated for and that there exist two solutions for each SOP within a unit cell of the two phase shifts.

Some key results have been illustrated together with important examples. The construction of intensity surfaces as a function of the two phase shifts depends on the input SOP. It is 2π periodic in both directions, its shape depends on the Stokes parameter $s_{1}$ only, while the phase shift δ between the two components of the Jones vector only causes a shift of the surface along the $\phi_{1}$ direction.

Conversely, it is proven that, in the controller operation, the device can generate any output SOP and that the effect of the $\phi_{1}$ and $\phi_{2}$ phase shifts is to trace parallels and meridians on Poincaré sphere, with respect to the $s_{1}$ axis. The rotation axis and angle are found as well.

An implicit equation for the 3 dB bandwidth, depending on the SOP and the particular choice of phase shifts pair, is derived. In general, the spectrum is not periodic and depends on both the SOP and the chosen solution. The effect on bandwidth of non-ideal factors resulting in unbalanced top and bottom phase shifter arms is studied and is proven not to be necessarily detrimental.

The curves traced with varying wavelength on Poincaré sphere by the SOP from the generator are found to be Clélie. More importantly, polarisations generated with the same $\phi_{2_{0}}/\phi_{1_{0}}$ ratio have been found to lie on the same Clélie moving along it with wavelength. The effect of unbalance is just to change the parameter identifying the particular Clélie and to rotate it by a fixed angle about the $s_{1}$ axis.

This geometrical representation and the mathematical analysis of this device illustrate the power of this approach and open the route to the analysis of more complex structures as well as useful treatment of such components in the quantum domain due to the direct correspondence of the Stokes parameters and the photon density operator.

## Figures and Tables

**Figure 1 micromachines-10-00364-f001:**
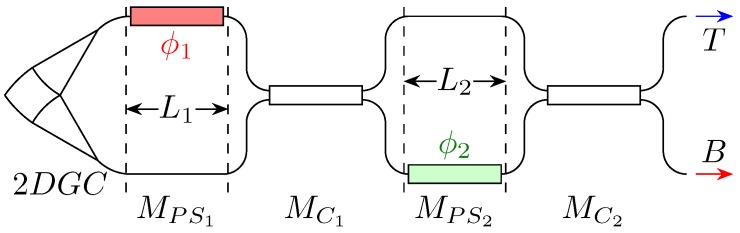
Schematic of the integrated polarisation controller. The labels $M_{i}$ refer to the transfer matrix associated with a given circuit section. The block on the picture leftmost portion is a schematic of the 2DGC. The “T” and “B” labels denote the top and bottom outputs, respectively.

**Figure 2 micromachines-10-00364-f002:**
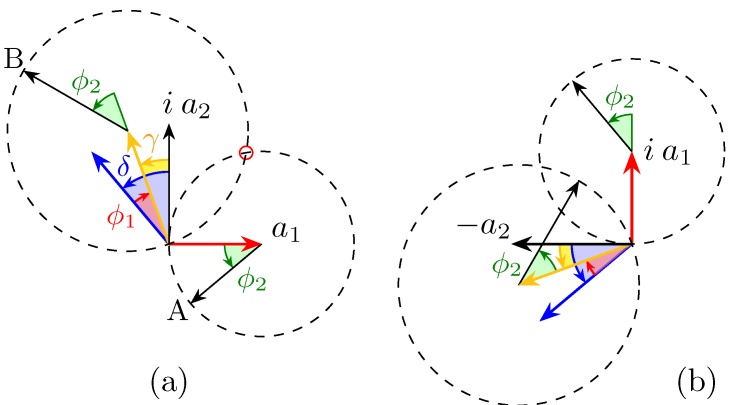
Phasor diagrams for the two components of $A_{out}$; actually, it shows the difference, not the sum, of the two terms of each component, as it is easier to visualise. (**a**) Top component of Equation (4). When the tips of the two vectors lie in the second intersection, circled in red, between the circumferences, then the first component of $A_{out}$ is zero. (**b**) Bottom component.

**Figure 3 micromachines-10-00364-f003:**
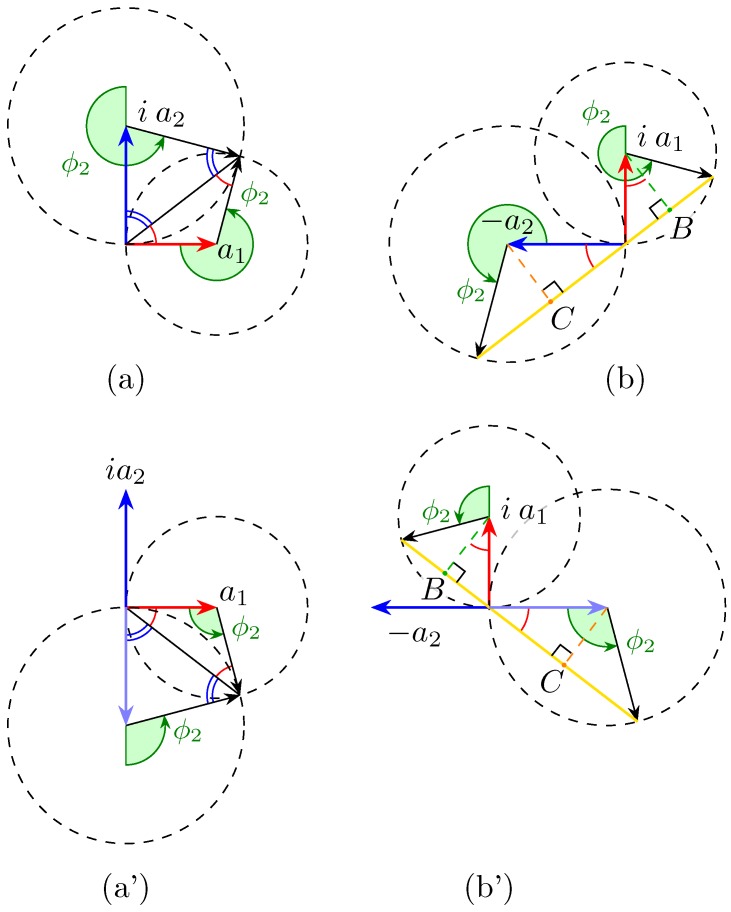
Phasor diagrams for the two components of $A_{\mathrm{out}}$, when both conditions of Equation (6) and (7) are fulfilled. (**a**) The four vectors form a quadrilateral. Note that Equation (7) can be deduced observing that the quadrilateral is the union of two congruent rectangle triangles with catheti $a_{1}$ and $a_{2}$ and thus the green angle equals twice $\arctan(a_{2}/a_{1})$ (or the explementary angle, for a’, when n is odd). (**b**) When Equation (6) is fulfilled, then the segments joining the origin with the points on the circumferences lie on the same line. The construction in figure shows that, when Equation (7) holds too, the segment between those two points is twice the hypotenuse of the rectangle triangle with $a_{1}$ and $a_{2}$ as catheti, which corresponds to the norm (power) of the input vector. Notice that, if one of the two angles (but not both) is changed by π, then the role in (a) and (b) gets reversed, i.e., the whole power goes in the top port.

**Figure 4 micromachines-10-00364-f004:**
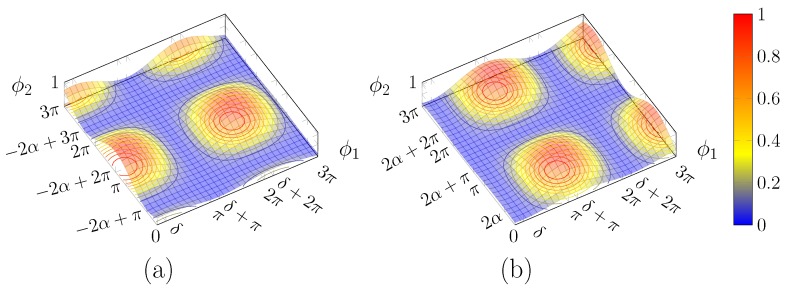
Plot of (**a**) $E(\phi_{1},\phi_{2})$ and (**b**) $O(\phi_{1},\phi_{2})$.

**Figure 5 micromachines-10-00364-f005:**
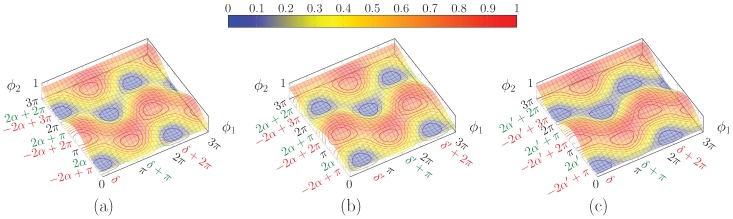
Plot of $P_{L}\left(\phi_{1},\phi_{2}\right)$ for: (**a**) α=π/3 and δ=π/6; (**b**) same α but δ=π/2, where the surface is shifted along $\phi_{1}$, but retains the same shape as (**a**); and (**c**) α=3/8π but same δ as in (**a**). Note that the surface has a different shape with respect to (**a**), even though the extrema lie in the same $\phi_{1}$ values. Red and green ticks correspond to the first and second solution classes, respectively.

**Figure 6 micromachines-10-00364-f006:**
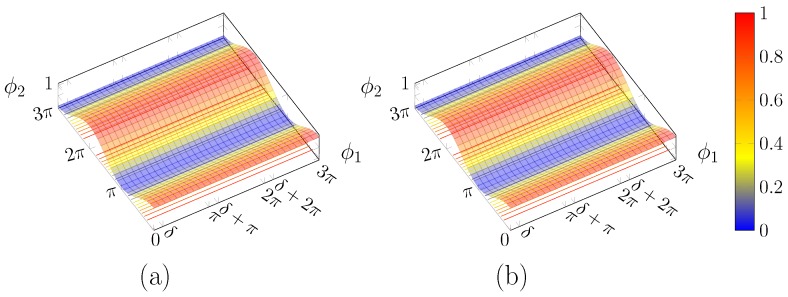
Plot of $P_{L}\left(\phi_{1},\phi_{2}\right)$ for: (**a**) α=0; and (**b**) α=π/2.

**Figure 7 micromachines-10-00364-f007:**
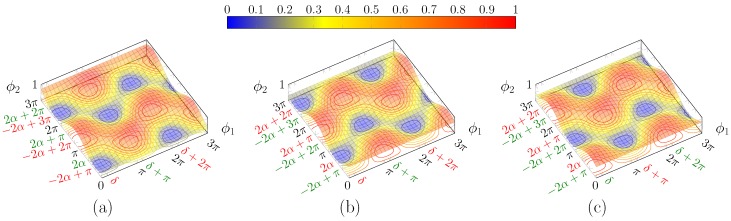
Plot of $P_{L}\left(\phi_{1},\phi_{2}\right)$ for: (**a**) α=π/3 and δ=π/6; (**b**) $\alpha^{\prime }=\pi /2-\alpha$ and same δ; and (**c**) $\alpha^{\prime }=\pi /2-\alpha$ and $\delta^{\prime }=\delta +\pi$. This is the complementary surface of (**a**). Note that the two solution classes are now inverted.

**Figure 8 micromachines-10-00364-f008:**
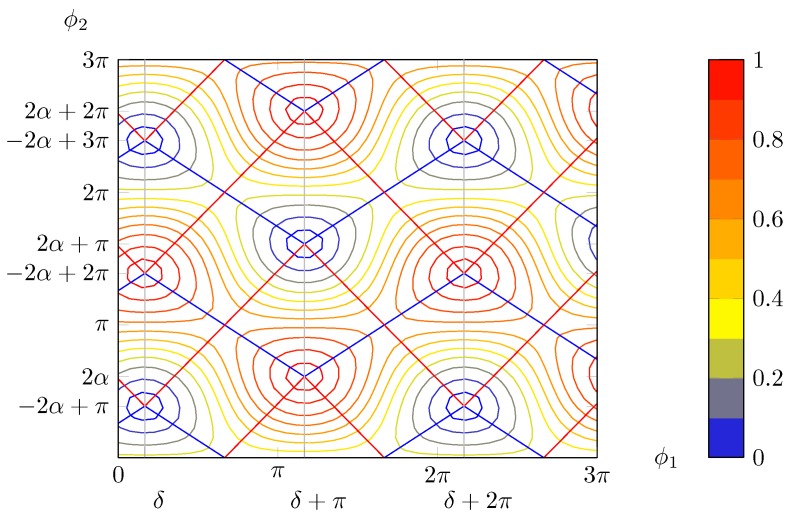
Contour plot for α=55° and δ=30°. In general, maxima and minima are not placed on a square grid. Blue lines join maxima lying on the same “ridge” or minima in the same “valley”, whereas red lines connect maxima separated by a “valley” or minima with a “ridge” in between, respectively.

**Figure 9 micromachines-10-00364-f009:**
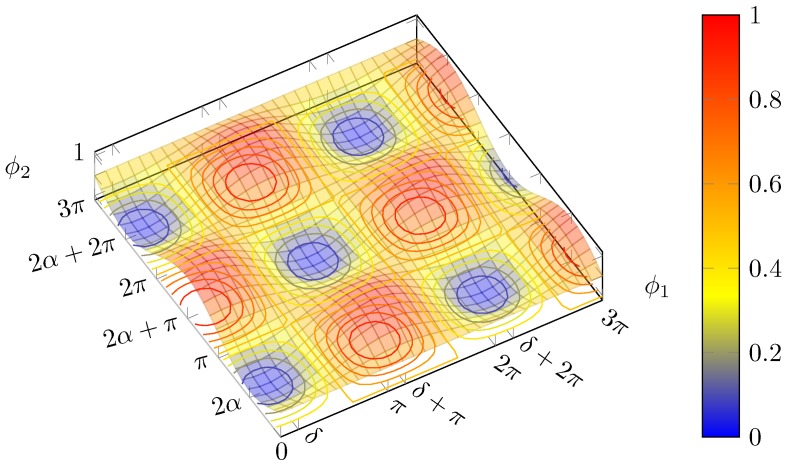
Plot of $P_{L}\left(\phi_{1},\phi_{2}\right)$ for α=π/4 and δ=π/6. Note that now the surface displays a check pattern.

**Figure 10 micromachines-10-00364-f010:**
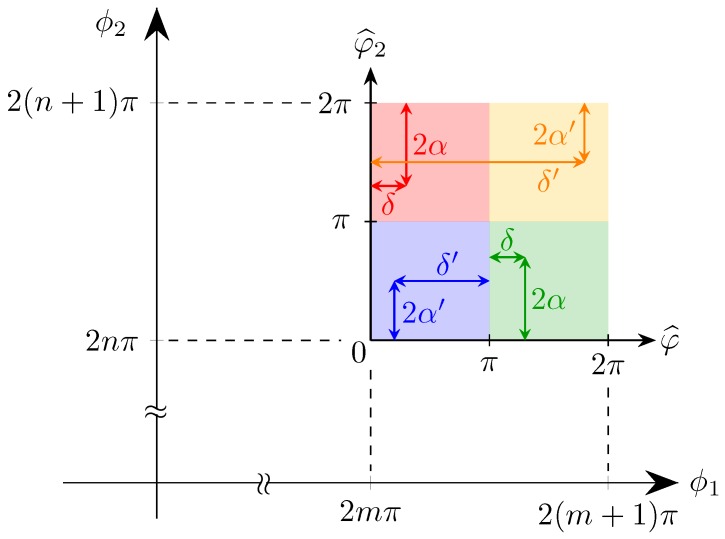
Diagram for the determination of the input SOP: $\phi_{1}$ and $\phi_{2}$ are the actual applied phase shifts, whereas $\hat{\varphi_{1}}$ and $\hat{\varphi_{2}}$ are the remainders; the new origin is placed in (2mπ,2nπ). The top quadrants (shaded in red and yellow) correspond to the first solution, $\left(\phi_{1}^{\prime },\phi_{2}^{\prime }\right)$, while the lower ones (blue and green) to the second solution. Polarisations with the same value of δ lie in diametrically opposed quadrants: the red and green concern the case of δ∈(0,π), whilst the yellow and blue ones to δ∈(π,2π).

**Figure 11 micromachines-10-00364-f011:**
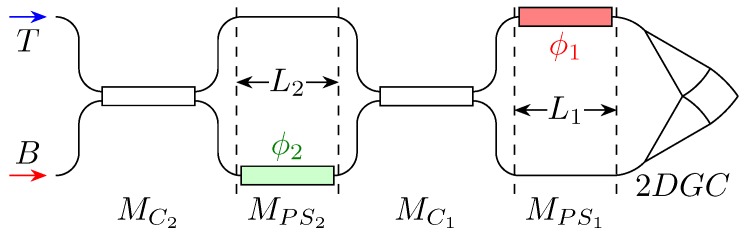
Schematic of the integrated polarisation controller. The labels are as in Figure 1.

**Figure 12 micromachines-10-00364-f012:**
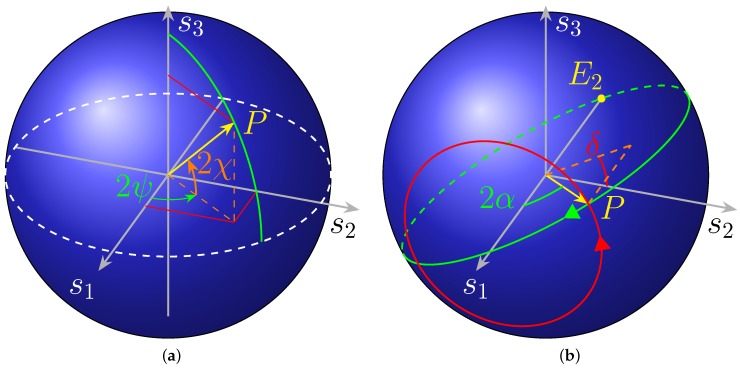
Poincaré sphere. (**a**) Any SOP can be represented by a point P on its surface, by Equation (42). The usual spherical coordinates are used. (**b**) Poincaré sphere with the angles as in Equation (41). The point *P* corresponds to a generic polarisation state. The red and green circles display the trajectory on the sphere when a complete sweep is performed on the angle δ ($\phi_{1}$) and α ($\phi_{2}$), respectively, while the other is held constant. Notice the direction of the arrows on the circles: scanning on δ results in a counterclockwise rotation, whereas the opposite happens for α, as the phase shifter is assumed to be in the lower arm.

**Figure 13 micromachines-10-00364-f013:**
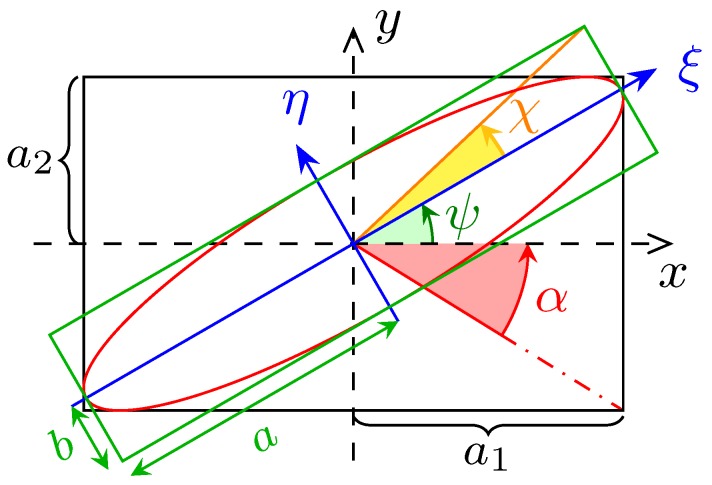
Elliptically polarised wave seen in the observer (xOy) and in the ellipse (ξOη) frame. The vibrational ellipse is for the electric field. The semi-major axis ξ is tilted from *x* by the angle ψ. The semi- axes length are *a* and *b*, not to be confused with the amplitudes $a_{1}$ and $a_{2}$ of the oscillations expressed in xOy.

**Figure 14 micromachines-10-00364-f014:**
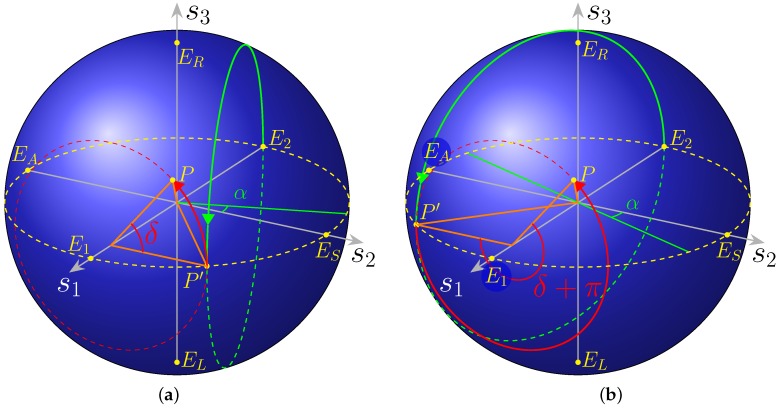
Action on SOP of the circuit, seen as a MZI followed by a phase shifter, for 2α=40°,δ=50°, in the case of (**a**) the first solution set; and (**b**) the second solution set.

**Figure 15 micromachines-10-00364-f015:**
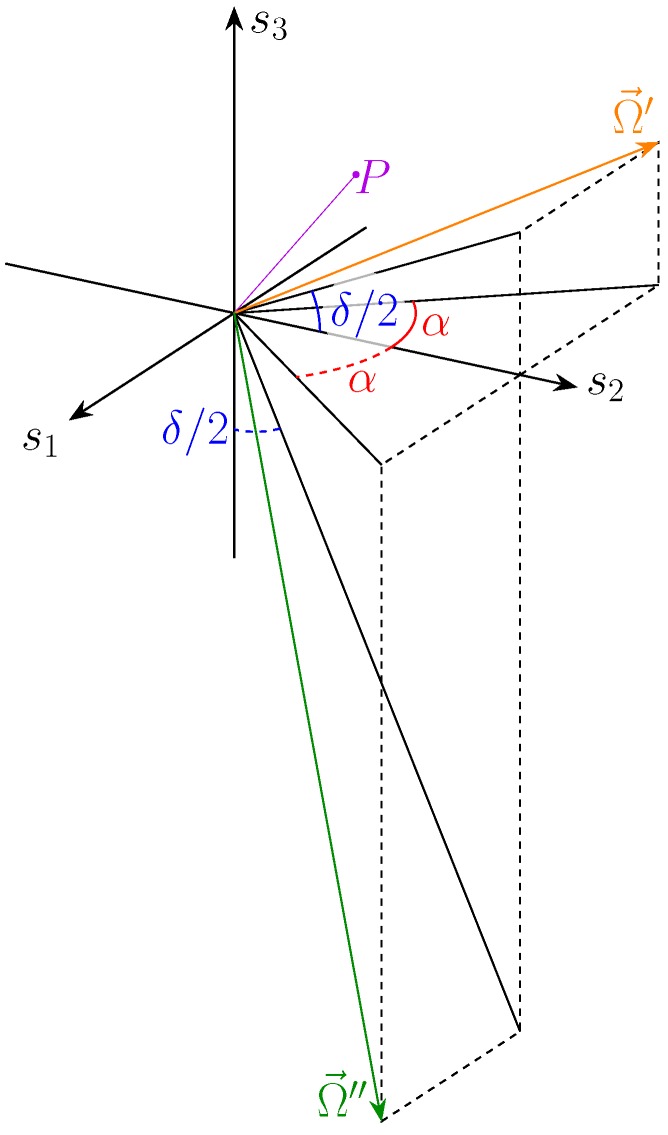
Rotation axes for the two solution sets.

**Figure 16 micromachines-10-00364-f016:**
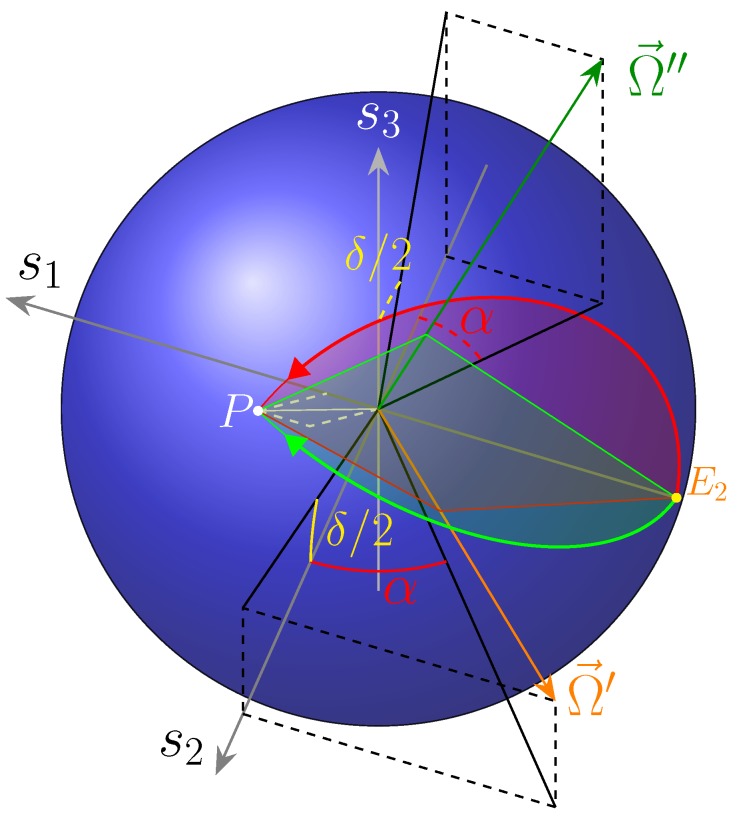
Overall device effect, for the two solutions.

**Figure 17 micromachines-10-00364-f017:**
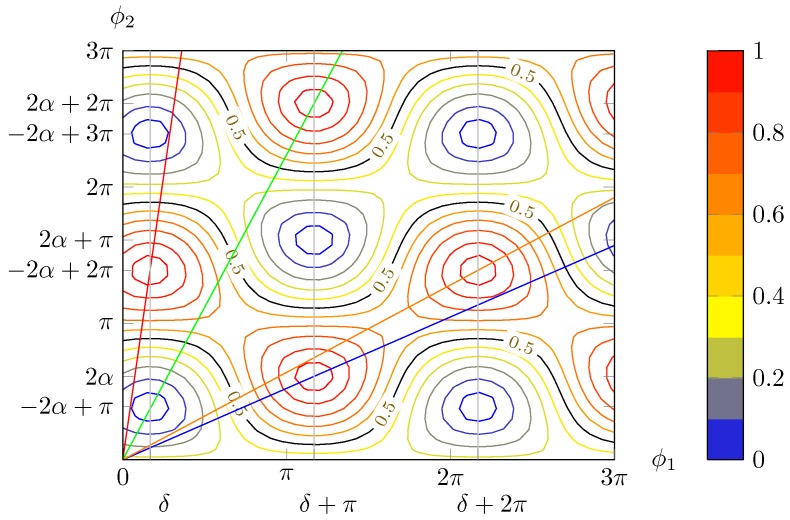
The pair of phase shifts as a function of frequency are lines (in different colours for each solution, refer to the legend of Figure 18) passing by the origin and the chosen peak, in the plane $\left(\phi_{1},\phi_{2}\right)$. For comparison, the contour plots for α=55° and δ=30° are included. Note that those lines in general do not pass through other maxima.

**Figure 18 micromachines-10-00364-f018:**
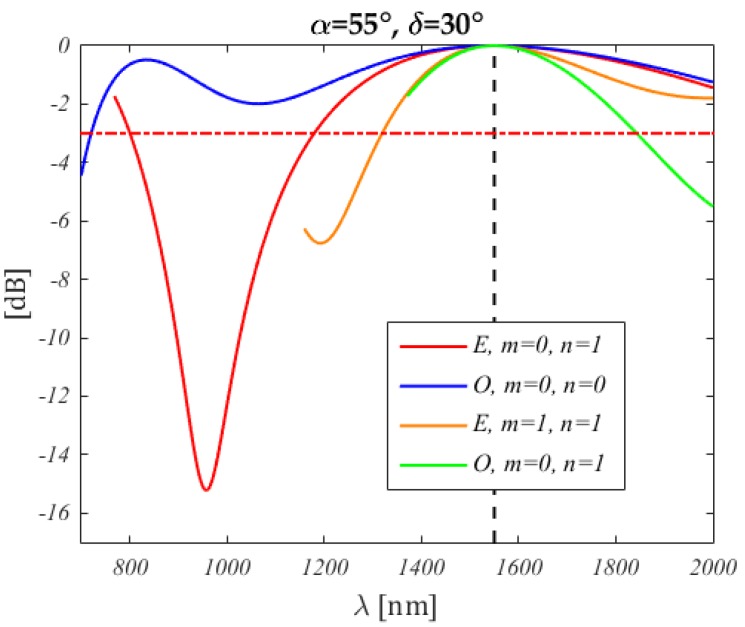
Spectra corresponding to the lines in Figure 17 (the colours correspond), for $f_{0}=$ 1934 THz (1550 nm, indicated by the black dashed vertical line). The wavelength ranges have been chosen to represent only the portion of the $\left(\phi_{1},\phi_{2}\right)$ plane shown in Figure 17. Note that the upper wavelength limit is the same, whereas the lower one increases with the solution distance from the origin. In general, the spectra are neither symmetric nor periodic. The legend lists the solution class (even or odd) and their order. The fundamental even solution is not reported, as it lies in the lower $\left(\phi_{1},\phi_{2}\right)$ half plane.

**Figure 19 micromachines-10-00364-f019:**
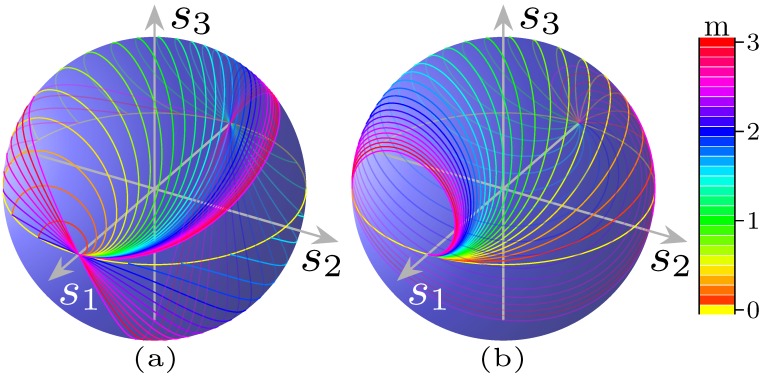
Family of Clélie curves, in the (**a**) direct form (Equation (91)), for the parameter ranging from m=0 to 3, at steps of 1/9. Note that curves with different values of the parameter may pass by the same point, corresponding to different order solutions that have a different frequency behaviour. (**b**) The inverse form (Equation (95)), for the same parameter values of (**a**), in growing order from red to violet. The equator, in yellow, corresponds to m=0, while in the previous figure it corresponds to m→∞.

**Figure 20 micromachines-10-00364-f020:**
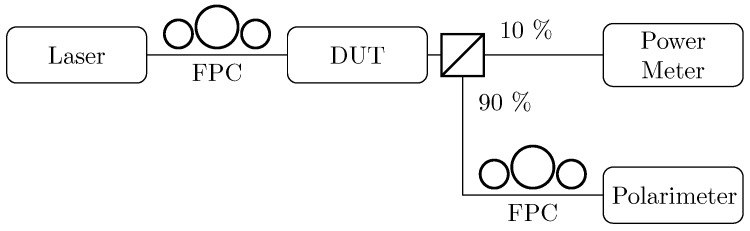
Setup used for the characterisation of the controller configuration. FPC, fibre polarisation controller; DUT, device under test.

**Figure 21 micromachines-10-00364-f021:**
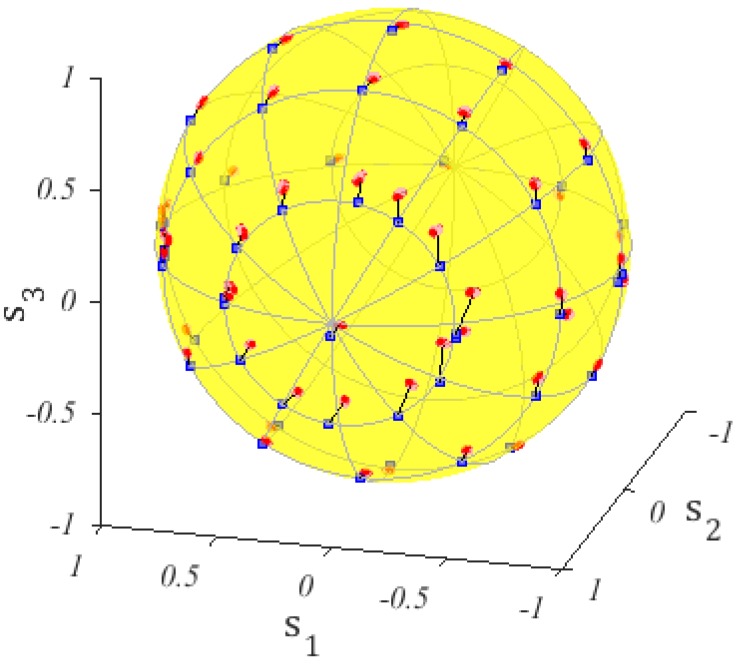
Comparison between experimental data points as read on the polarimeter (blue squares) and as deduced from the dissipated power on each heater (red), measured by the sourcemeter and correcting the effect of the parasitic common ground electrode resistance. The start point for $P_{1}$ and $P_{2}=0$ is that with $s_{1}=-1$. Increasing $P_{1}$ produces a counter clockwise rotation around the $s_{2}$ axis, $P_{2}$ on the $s_{3}$ axis, when P1=0.

**Figure 22 micromachines-10-00364-f022:**
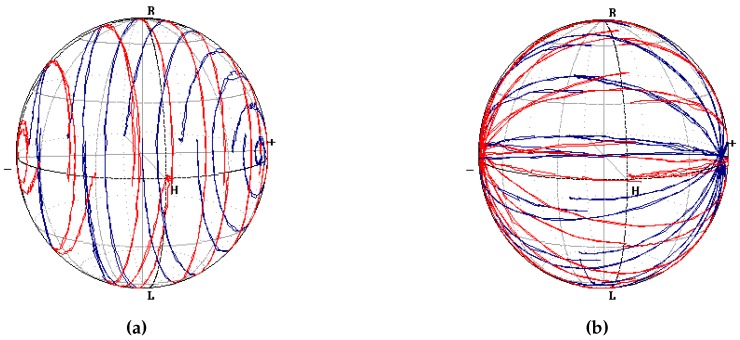
Curves traced on the polarimeter when scanning on (**a**) $\phi_{1}$ and (**b**) $\phi_{2}$, while keeping the other phase constant. The polarimeter has been calibrated so that the SOP from the SPGC is shown as a 45° linear polarisation. Red curves lie on the hemisphere on the observer’s side, blue ones on the rear one. The axes are rotated by 90° around $s_{3}$, with respect to Figure 21.

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
