# Peer review of "Geometrical Representation of a Polarisation Management Component on a SOI Platform"

_micromachines, 2019, doi:10.3390/mi10060364_

Round 1

Reviewer 1 Report

The work is studying polarization related component in silicon photonics, which has broad impact in the field. However, the manuscript does not introduce the field or motivate their work properly. In particular, the over-simplification of the butt coupling (where the claim is also not true) while ignoring many other coupling schemes in quite misleading to general readers. The approach they used is also not motivated sufficiently. Most importantly, the key results and logic are also overwhelmed by the equations and graphs.

Overall, in my point of view, this manuscript needs to be significantly improved in presentation before considering publication in scientific journals.

Reviewer 2 Report

This manuscript presents an analytical study of state of polarization in a silicon photonic circuit. The authors use a geometrical representation to develop an intuitive understanding of the operation of the photonic circuit. The manuscript may be published in the special issue of Micromachones provided the following issues are addressed in a revised manuscript:

line 25-26: this statement should be deleted. The photonic circuit analyzed has already been pubished [1,6,10].

line 37-50: this paragraph should be shortened. The endless properties of the circuit has already been shown in [10] and there is no need to elaborate on this point here.

line 78: this needs to be clarified. There is no photodetector shown in figure 1. 

Figure 1: the grating coupler should be identified in the figure. 

line 93: are there any required assumption regarding any polarization dependence in 3-dB couplers? 

Equation 4: it would be clearer if the condition on A_out was stated in a separate equation instead of the last identity in eqn 4.

Section 4. Characterization: This section is too brief. Enough details on the components and the experiment should be added for the reader to be able to repeat the measurements. Is the photonic circuit the same as used in [6]?

line 466: this sentence is unclear. 

line 470-473: this is also unclear/confusing and must be rewritten. 

Figure 21 and 22: the effect of the parasitic resistance should be elaborated. How is the “non-closing” of the traced in Figure 22 related to the errors in Fig 21? 

Section 2.1.2: These 4 pages could be reduced significantly without loosing any insight. 

Section 3.2: This section could be shortened significantly to improve readability. 

In general, the manuscript suffers from an excessive amount of details, reducing readability, contrary to the claimed contribution to “intuitive understanding”. This could be improved by rewriting. 

Reviewer 3 Report

In this manuscript, author derived the operation principle of a polarization control device on SOI template. The mathematical derivation are detailed and appreciated. The manuscript is recommended publish before some minor issues addressed.

·       A little bit quantitative estimation of polarization variation on PIC performance in practical situation strengthen the motivation.

·       In Fig. 1, what is the portion on the left most of the schematic?

·       On the top right, it is better to add a PD in the schematic.

·       On page 3 line 92-96, please validate the main hypothesis that the 2DGC can splite the TE and TM components without affecting their relative amplitude and phases. What kind of device can achieve this?

·       On page 5 line 127, why set the bottom equation of eqn. (4) to zero? It should be equal to exp(iθ).

·       On page 5 line 129, there is a typo. “tem” should be “term”.

·       Please add color scale to all of the contour plot.

·       On page 19 section 3, what does the “frequency” refer to? Does it refer to the speed at which this device can adjust the polarization?

·       On page 24 line 442, what does the bandwidth refer to? Does it refer to the wavelength range of the optical signal?

·       How was the device fabricated? Was it fabricated by commercial CMOS foundry?

·       For the characterization of the device, was the response speed (i.e. how fast the device can respond to the variation of the polarization) of the device measured?

·       There are lots of places where the first line of the paragraph should be hanging and other places the line should not be hanging. Please correct them. For example, line57, 62, 64, 127, 129,166,178, 185, 214, 223, 237, 255, 261, 311, 329, 341, 348, 349, 363, 376, 466, 487, 491, 498, etc.
